# New particle formation in the sulfuric acid-dimethylamine-water system: Reevaluation of CLOUD chamber measurements and comparison to an aerosol nucleation and growth model

Andreas Kürten[1], Chenxi Li[2], Federico Bianchi[3], Joachim Curtius[1], António Dias[4], Neil M. Donahue[5], Jonathan Duplissy[3], Richard C. Flagan[6], Jani Hakala[3], Tuija Jokinen[3], Jasper Kirkby[1,7], Markku Kulmala[3], Ari Laaksonen[8], Katrianne Lehtipalo[3,9], Vladimir Makhmutov[10], Antti Onnela[7], Matti P. Rissanen[3], Mario Simon[1], Mikko Sipilä[3], Yuri Stozhkov[10], Jasmin Tröstl[9], Penglin Ye[5,11], and Peter H. McMurry[2]

[1]Institute for Atmospheric and Environmental Sciences, Goethe University Frankfurt, 60438 Frankfurt am Main, Germany.

[2]Department of Mechanical Engineering, University of Minnesota, 111 Church St. SE, Minneapolis, MN 55455, USA.

[3]Department of Physics, University of Helsinki, FI-00014 Helsinki, Finland.

[4]SIM, University of Lisbon, 1849-016 Lisbon, Portugal.

[5]Center for Atmospheric Particle Studies, Carnegie Mellon University, Pittsburgh, Pennsylvania 15213, USA.

[6]Division of Chemistry and Chemical Engineering, California Institute of Technology, Pasadena, California 91125, USA.

[7]CERN, CH-1211 Geneva, Switzerland.

[8]Finnish Meteorological Institute, FI-00101 Helsinki, Finland.

[9]Laboratory of Atmospheric Chemistry, Paul Scherrer Institute, 5232 Villigen PSI, Switzerland.

[10]Solar and Cosmic Ray Research Laboratory, Lebedev Physical Institute, 119991 Moscow, Russia.

[11]Aerodyne Research Inc., Billerica, Massachusetts 01821, USA.

Correspondence to: Andreas Kürten (kuerten@iau.uni-frankfurt.de)

**Abstract**

A recent CLOUD (Cosmics Leaving OUtdoor Droplets) chamber study showed that sulfuric acid and dimethylamine produce new aerosols very efficiently, and yield particle formation rates that are compatible with boundary layer observations. These previously published new particle formation (NPF) rates are re-analyzed in the present study with an advanced method. The results show that the NPF rates at 1.7 nm are more than a factor of 10 faster than previously published due to earlier approximations in correcting particle measurements made at larger detection threshold. The revised NPF rates agree almost perfectly with calculated rates from a kinetic aerosol model at different sizes (1.7 nm and 4.3 nm mobility diameter). In addition, modeled and measured size distributions show good agreement over a wide range (up to ca. 30 nm). Furthermore, the aerosol model is modified such that evaporation rates for some clusters can be taken into account; these evaporation rates were previously published from a flow tube study. Using this model, the findings from the present study and the flow tube experiment can be brought into good agreement for the high base to acid ratios (~100) relevant for this study. This confirms that nucleation proceeds at rates that are compatible with collision-controlled (a.k.a. kinetically-controlled) new particle formation for the conditions during the CLOUD7 experiment (278 K, 38% RH, sulfuric acid concentration between $1 \times 10^6$ and $3 \times 10^7$ cm$^{-3}$ and dimethylamine mixing ratio of ~40 pptv, i.e., $1 \times 10^9$ cm$^{-3}$).

## 1. INTRODUCTION

The formation of new particles by gas-to-particle conversion (nucleation or new particle formation, NPF) is important for a variety of atmospheric processes and for human health.

It has been shown in numerous studies that sulfuric acid ($H_2SO_4$) is often associated with NPF (Weber et al., 1997; Kulmala et al., 2004; Fiedler et al., 2005; Kuang et al., 2008; Kirkby et al., 2011) and indeed it can explain some of the observed particle formation together with water vapor for neutral (uncharged) and ion-induced conditions when temperatures are low, e.g., in the free troposphere (Lee et al., 2003; Lovejoy et al., 2004; Duplissy et al., 2016; Ehrhart et al., 2016; Dunne et al., 2016). However, at least one additional stabilizing compound is required in order to explain boundary layer nucleation at warm temperatures. Acid-base nucleation, which involves a ternary compound, e.g., ammonia, besides sulfuric acid and water, can lead to much higher NPF rates compared to the binary system (Weber et al., 1998; Ball et al., 1999; Kürten et al., 2016a). Nevertheless, for most conditions close to the surface, the concentrations of $H_2SO_4$ and $NH_3$ are too low, or temperatures are too high, to allow significant ternary nucleation of these compounds (Kirkby et al., 2011; Kürten et al., 2016a). However, the substitution of ammonia by amines, e.g., dimethylamine ($(CH_3)_2NH$), leads to NPF rates that can explain the atmospheric observations over a wide range of sulfuric acid concentrations, even when the amine mixing ratios are in the low pptv-range (Kurtén et al., 2008; Nadykto et al., 2011; Ortega et al., 2012; Chen et al., 2012; Almeida et al., 2013; Glasoe et al., 2015). A recent study even showed that NPF is collision-controlled, i.e., that it proceeds at the maximum possible speed (Rao and McMurry, 1989), when amine mixing ratios are above ~20 pptv ($5 \times 10^8$ $cm^{-3}$), and sulfuric acid concentrations are between $1 \times 10^6$ $cm^{-3}$ and $3 \times 10^7$ $cm^{-3}$ at 278 K and 38% RH (Kürten et al., 2014). Indications that NPF can be collision-limited were reported more than 30 years ago based on the analysis of chamber nucleation experiments (McMurry, 1980), although the involvement of amines, which were probably present as a contaminant during those experiments, was not considered. Indications that atmospheric nucleation might occur by a collision-limited process have also been previously presented (Weber et al., 1996). Despite the strong evidence that sulfuric acid-amine nucleation is very efficient, it has rarely been observed in the atmosphere. Only one study has so far reported sulfuric acid-amine nucleation (Zhao et al., 2011) despite amine mixing ratios of up to tens of pptv at some sites (Yu and Lee, 2012; You et al., 2014; Freshour et al., 2014; Yao et al., 2016). A global modelling study of sulfuric acid-amine nucleation has been carried out so far (Bergman et al., 2015) applying a nucleation parametrization based on the measurements of Almeida et al. (2013) and Glasoe et al. (2015).

Atmospheric boundary layer nucleation can also be explained by the existence of highly-oxygenated organic molecules (Crounse et al., 2013; Ehn et al., 2014), e.g., from α-pinene. These highly-oxygenated molecules have been found to nucleate efficiently in a chamber study even without the involvement of sulfuric acid, especially when ions take part in the nucleation process (Kirkby et al., 2016).

Even though oxidized organics seem to be globally important for NPF (Jokinen et al., 2015; Gordon et al., 2016; Dunne et al., 2016), the formation of new particles by sulfuric acid and amines should still be considered because sulfuric acid-amine nucleation rates exceed those from oxidized organics as soon as the concentrations of the precursor gases (sulfuric acid and

amines) are high enough (Berndt et al., 2014). Therefore, at least locally or regionally, i.e., close
to sources, amines should be relevant.
In this study, we reanalyze data from CLOUD (Cosmics Leaving OUtdoor Droplets)
chamber experiments conducted at CERN during October/November 2012 (CLOUD7
campaign). New particle formation rates as a function of the sulfuric acid concentration from
CLOUD7 were previously published (Almeida et al., 2013). However, these data are re-
analyzed in the present study using an advanced method that takes into account the effect of
self-coagulation in the estimation of new particle formation rates (Kürten et al., 2015a). The re-
analyzed data and NPF rates obtained from Scanning Mobility Particle Sizer (SMPS)
measurements are compared to results from a kinetic aerosol model. Modeling is also used for
a comparison between results from a flow tube study (Jen et al., 2016a) and CLOUD.
The reanalyzed data cover sulfuric acid concentrations from ca. $1 \times 10^6$ to $3 \times 10^7$ cm$^{-3}$, which
fall into the range for most observations of atmospheric boundary layer new particle formation
events (e.g. Kulmala et al., 2013). The dimethylamine mixing ratio for most of the data shown
in this study is ~40 pptv ($1 \times 10^9$ cm$^{-3}$), which is within the rather wide range of observations
(0.1 to 157 pptv, i.e., $2.5 \times 10^6$ to $4 \times 10^9$ cm$^{-3}$) for C2-amines to which dimethylamine belongs to
(Yao et al., 2016).
**2. METHODS**
**2.1 CLOUD experiment and instruments**
The CLOUD (Cosmics Leaving OUtdoor Droplets) experiment at CERN was designed to
investigate nucleation and growth of aerosol particles in chemically diverse systems.
Additionally, the influence of ions on new particle formation (NPF) and growth can be studied
inside the 26.1 m$^3$ electro-polished stainless steel chamber (Kirkby et al., 2011). For the
experiments discussed in this paper, NPF is initiated by illuminating the air inside the chamber
with UV light by means of a fiber-optic system (Kupc et al., 2011), which produces sulfuric
acid ($H_2SO_4$) photolytically from reactions involving $O_3$, $H_2O$, $SO_2$ and $O_2$. Diluted
dimethylamine and sulfur dioxide are taken from gas bottles; inside the chamber, these trace
gases mix with clean synthetic air (i.e., $O_2$ and $N_2$ with a ratio of 21:79 from evaporated
cryogenic liquids). To ensure homogenous conditions, the air is mixed with magnetically driven
fans installed at the top and bottom of the chamber (Voigtländer et al., 2012). A thermal housing
controls the chamber temperature to 278.15 K within several hundredths of a degree. The
temperature was not varied for the experiments relevant for this study. The relative humidity
was kept constant at 38% by humidifying a fraction of the inflowing air with a humidification
system (Duplissy et al., 2016). In order to keep the pressure inside the chamber at 1.005 bar,
the air that is taken by the instruments has to be continuously replenished. Therefore, a flow of
150 l/min of the humidified air is continuously supplied to the chamber. For the sulfuric acid,
dimethylamine and water system, ions do not have a strong enhancing effect on the nucleation
rates for most conditions (Almeida et al., 2013); therefore, we do not distinguish between the
neutral and charged pathway in such runs.

A suite of instruments is connected to the CLOUD chamber to measure particles, ions, clusters and gas concentrations. A summary of these instruments is provided elsewhere (Kirkby et al., 2011; Duplissy et al., 2016). For this study, measured sulfuric acid and particle concentrations are relevant. A Chemical Ionization-Atmospheric Pressure interface-Time Of Flight Mass Spectrometer (CI-APi-TOF) was employed to measure sulfuric acid and its neutral clusters in this study (Jokinen et al., 2012; Kürten et al., 2014). The particle concentrations originate from a scanning mobility particle sizer (SMPS, Wang and Flagan, 1990), which measured the particle size distribution between ~4 and ~80 nm. The SMPS uses a differential mobility analyzer built by the Paul Scherrer Institute; it includes a Kr[85] charger to bring the particles into a charge equilibrium before they are classified. The retrieval of the particle size distributions requires corrections for the charging and the transmission efficiency, which were performed according to the literature (Wiedensohler and Fissan, 1988; Karlsson and Martinsson, 2003). The mixing ratio of dimethylamine was determined by ion chromatography with a detection limit of 0.2 to 1 pptv ($5\times10^6$ to $2.5\times10^7$ cm$^{-3}$) at a time resolution between 70 and 210 minutes (Praplan et al., 2012; Simon et al., 2016).

## 2.2 Calculation of particle formation rates

Particle formation rates $J$ (cm$^{-3}$ s$^{-1}$) are calculated from the measured size distributions (assumed to consist of $n$ bins). For the size bin with the index $m$, the rate at which particles with a diameter equal or larger than $d_m$ are formed can be calculated according to Kürten et al., 2015a:

$$J_{\geq m} = \frac{dN_{\geq m}}{dt} + \sum_{i=m}^{n}(k_{w,i} \cdot N_i) + k_{dil} \cdot N_{\geq m} + \sum_{i=m}^{n}\left(\sum_{j=i}^{n} s_{i,j} \cdot K_{i,j} \cdot N_j \cdot N_i\right). \qquad (1)$$

This equation takes into account the time derivative of the number density of all particles for which $d_p \geq d_m$, i.e., $N_{\geq m}$, and corrects for the effects of wall loss (size dependent wall loss rates $k_{w,i}$), dilution (dilution rate $k_{dil}$), and coagulation (collision frequency function $K_{i,j}$), where $N_i$ and $N_j$ are the particle number densities in different size bins. The rate of losses to the chamber walls can be expressed by Crump and Seinfeld, 1981:

$$k_w(d_p) = C_w \cdot \sqrt{D(d_p)}, \qquad (2)$$

where $D(d_p)$ is the diffusivity of a particle of diameter $d_p$, which is given by the Stokes-Einstein relation (Hinds, 1999),

$$D(d_p) = \frac{k_B \cdot T \cdot C_C}{3 \cdot \pi \cdot \eta \cdot d_p}, \qquad (3)$$

where $k_b$, $T$, $\eta$, are the Boltzmann constant, the temperature, and the gas viscosity, respectively. The Cunningham slip correction factor, $C_C$, is a function of the particle Knudsen number, $Kn = 2\lambda/d_p$, and $\lambda$ is the mean-free-path of the gas molecules. The empirically derived proportionality coefficient, $C_w$, depends upon the chamber dimensions and on the intensity of turbulent mixing.

The rate of loss of sulfuric acid to the chamber walls is generally used to characterize $C_w$. The diffusivity of sulfuric acid is 0.0732 cm$^2$ s$^{-1}$ at 278 K and 38% RH (Hanson and Eisele, 2000). The measured life time, determined from the decay of sulfuric acid when the UV light is turned off, was 554 s (wall loss rate 0.00181 s$^{-1}$), with the experimentally determined diffusivity this yields a factor $C_w$ of 0.00667 cm$^{-1}$ s$^{-0.5}$. However, in this study diffusivities were calculated according to equation (3), so the calculated monomer diffusivity (for a monomer with a density of 1470 kg m$^{-3}$ and a molecular weight of 0.143 kg mol$^{-1}$, see section 2.4) required a different scaling, resulting in a value of $C_w = 0.00542$ cm$^{-1}$ s$^{-0.5}$ that was used throughout this study.

Dilution is taken into account by a loss rate that is independent of size and equals $k_{dil} = 9.6 \times 10^{-5}$ s$^{-1}$. Correcting for particle-particle collisions requires the calculation of the collision frequency function. We used the method from Chan and Mozurkewich (2001). This method includes the effect of enhanced collision rates through van der Waals forces. A value of $6.4 \times 10^{-20}$ J was used for the Hamaker constant (Hamaker, 1937), leading to a maximum enhancement factor of ~2.3 for the smallest clusters, relative to the collision rate in the absence of van der Waals forces. The factor of 2.3 has previously been shown to give good agreement between measured and modeled cluster and particle concentrations for the chemical system of sulfuric acid and dimethylamine (Kürten et al., 2014; Lehtipalo et al., 2016). In order to consider the collisions of particles in the same size bin, a scaling factor $s_{i,j}$ is used in equation (1), which is 0.5 when $i = j$ and 1 otherwise.

**2.3 Reconstruction method**

Recently a new method was introduced, that makes it possible to retrieve new particle formation rates at sizes below the threshold of the instrument used to determine the particle number density. This method is capable of considering the effect of self-coagulation (Kürten et al., 2015a). It requires introducing new size bins below the threshold of the SMPS (termed $d_{p2}$ in the following; $d_{p2}$ corresponds to the index $m = 1$). The method starts by calculating the number density in the first newly introduced smaller size bin (index $m = 0$, diameter $d_{p2}$ - d$d_p$):

$$N_{m-1} = \left(d_{p,m} - d_{p,m-1}\right) \cdot \frac{J_{\geq m}}{GR_{m-1}} \approx \mathrm{d}d_p \cdot \frac{J_{\geq m}}{GR}. \qquad (4)$$

Here, the particle growth rate $GR$ (nm s$^{-1}$) needs to be used as well as the difference between two adjacent size bins (d$d_p$). Once the number density in the newly introduced bin is known this information can be used to calculate $J_{m-1}$. In the further steps, the numbers $N_{m-2}$ and $J_{m-2}$ are calculated and so on. In this way, the size distribution can be extrapolated towards smaller and smaller sizes in a stepwise process until eventually reaching the diameter $d_{p1}$.

The method has so far only been tested against simulated data but not against measured size distributions (Kürten et al., 2015a). In this study the smallest measured SMPS diameter is $d_{p2} = 4.3$ nm; 26 new size bins with d$d_p = 0.1$ nm were introduced and this enabled the calculation of the NPF rates at $d_{p1} = 1.7$ nm in the smallest size bin. This size was chosen since previously published particle formation rates from the CLOUD experiment were reported for this diameter (e.g. Kirkby et al., 2011; Almeida et al., 2013; Riccobono et al., 2014).

The method introduced here explicitly takes into account losses that occur between particles with $d_{p1}$ and $d_{p2}$ (self-coagulation). These losses have not been taken into account by Almeida

et al. (2013). Almeida et al. (2013) derived $J_{3.2nm}$ from CPC and SMPS measurements by
including the corrections for wall loss, dilution and coagulation above 3.2 nm (see also Kürten
et al., 2016a). However, the extrapolation to 1.7 nm was made by using the Kerminen and
Kulmala equation (Kerminen and Kulmala, 2002), which does not include the effect of self-
coagulation. For the system of sulfuric acid and dimethylamine, where a significant fraction of
particles reside in the small size range, this process is, however, important.
**2.4 Kinetic new particle formation and growth model**
The measured particle formation rates are compared to modeled formation rates assuming
collision-limited particle formation, i.e., all clusters are not allowed to evaporate. McMurry
(1980) was the first to show that number concentrations and size distributions of particles
formed photochemically from $SO_2$ in chamber experiments (Clark and Whitby, 1975) are
consistent with collision-controlled nucleation; results from updated versions of this model
have recently been presented (Kürten et al., 2014; McMurry and Li, 2017). The model used
here has been described previously (Kürten et al., 2014; Kürten et al., 2015a, Kürten et al.
2015b) but only brief introductions were reported; therefore, more details are provided in the
following.
As outlined in Kürten et al. (2014), collision-controlled new particle formation accurately
described the measured cluster distributions for the sulfuric acid-dimethylamine system up to
the pentamer (cluster containing five sulfuric acid molecules). In this model, it was assumed
that the clusters consist of "monomeric" building blocks, each containing one dimethylamine
and one sulfuric acid molecule. Evidence that this 1:1-ratio between base and acid is
approximately maintained for the small clusters was presented from neutral and charged cluster
measurements (Almeida et al., 2013; Kürten et al., 2014; Bianchi et al., 2014; Glasoe et al.,
2015). The molecular weight was, therefore, chosen as 0.143 kg mol$^{-1}$ (sum of sulfuric acid
with 0.098 kg mol$^{-1}$ and dimethylamine with 0.045 kg mol$^{-1}$), and the density as 1470 kg m$^{-3}$
(Qiu and Zhang, 2012).
During the reported experiments (CLOUD7 in fall 2012), dimethylamine was always present
at mixing ratios above ca. 20 pptv ($5\times10^8$ cm$^{-3}$). Dimethylamine (DMA) was supplied from a
certified gas bottle and diluted with synthetic air before it was introduced into the chamber to
achieve the desired mixing ratios. Sulfuric acid was generated in situ from the reactions between
$SO_2$ and OH whenever the UV light was turned on (see section 2.1). Since the UV light intensity
and the gas concentrations were kept constant throughout each individual experiment, it is
justified to assume a constant monomer production rate $P_1$. The equation describing the
temporal development of the monomer concentration, $N_1$, is
$$\frac{dN_1}{dt} = P_1 - \left(k_{1,w} + k_{dil} + \sum_{j=1}^{N_{max}} K_{1,j} \cdot N_j\right) \cdot N_1 \qquad (5)$$
and, for the clusters containing two or more sulfuric acid molecules ($k \geq 2$),
$$\frac{dN_k}{dt} = \frac{1}{2} \cdot \sum_{i+j=k} K_{i,j} \cdot N_i \cdot N_j - \left(k_{w,k} + k_{dil} + \sum_{j=1}^{N} K_{k,j} \cdot N_j\right) \cdot N_k. \qquad (6)$$

The same loss mechanisms (wall loss, dilution and coagulation) as for the calculation of the
particle formation rates (section 2.2) are considered when modeling the cluster concentrations.
In this study, the particle size distribution was calculated from the monomer up to a diameter
of ~84 nm, which corresponds to the upper size limit of the SMPS used in CLOUD7. Tracking
each individual cluster/particle up to this large size would be computationally too demanding,
so the size distribution was divided into so-called molecular size bins (tracking each individual
cluster), and geometric size bins, where the mid-point diameters of two neighboring size bins
differ by a constant factor. The number of molecular size bins was set to 400 (which results in
a diameter of ~5 nm for the largest molecular bin), while the number of geometric size bins was
set to 190 with a geometric factor of 1.015 (maximum diameter of the last bin is 83.7 nm). The
treatment of the geometric size bins was similar to the molecular bins, except that the collision
products were distributed between the two closest size bins. Two smaller particles with
diameters $d_{p,i}$ and $d_{p,j}$ generate a cluster with size

$$d_{p,x} = \left( d_{p,i}^3 + d_{p,j}^3 \right)^{1/3}. \tag{7}$$

If it is assumed that the collision product falls into the size range covered by the geometric bins,
its diameter will be between two size bins $d_{p,k}$ and $d_{p,k+1}$. The production rate of particles with
diameter $d_{p,x}$ is

$$P_x = s_{i,j} \cdot K_{i,j} \cdot N_i \cdot N_j. \tag{8}$$

For the geometric size range, the resulting particles are distributed between the two bins to
conserve mass, i.e.,

$$P_k = \left( \frac{d_{p,k+1}^3 - d_{p,x}^3}{d_{p,k+1}^3 - d_{p,k}^3} \right) \cdot P_x, \tag{9a}$$
$$P_{k+1} = \left( 1 - \frac{d_{p,k+1}^3 - d_{p,x}^3}{d_{p,k+1}^3 - d_{p,k}^3} \right) \cdot P_x. \tag{9b}$$

When the collision product falls into the molecular size bin regime the calculation is
straightforward because the diameter of the product agrees exactly with a molecular bin and
does not need to be distributed between two bins (see the production term in equation (6)). In
case the collision products exceed the largest bin diameter, the product is entirely assigned to
the largest bin, while taking into account the scaling such that the total mass is conserved.
In the model, no free parameter is used as the concentration of monomers is constrained by
the measurements. Therefore, the production rate $P_1$ is adjusted such that the resulting monomer
concentration in the model matches the measured sulfuric acid concentration. The model is used
to simulate the experiments for a duration of 10,000 s with a time resolution of 1 s. For the
small clusters and particles this leads to a steady-state between production and loss; therefore,
the resulting concentrations are essentially time-independent.
The model introduced here was compared with the model described in McMurry and Li
(2017) and yielded almost indistinguishable results for several scenarios when the same input
parameters were used. We take this as an indication that both models correctly describe

collision-controlled nucleation, especially since the models were independently developed and do not share the same code. The model in this paper is based on defining size bins according to their diameter, while the model by McMurry and Li (2017) uses particle volume.

**2.5 Nucleation and growth model involving selected evaporation rates**

Measured cluster concentrations for the sulfuric acid-dimethylamine system from flow tube experiments indicated that finite evaporation rates exist for some clusters (Jen et al., 2014; Jen et al., 2016a). This was supported by the observation that diamines can yield even higher formation rates than amines for some conditions (Jen et al., 2016b). Within the flow tube experiments dimethylamine was mixed into a gas flow containing a known amount of sulfuric acid monomers. The products, i.e., the sulfuric acid-dimethylamine clusters were measured after a short reaction time ($\leq$ 20 s) with a chemical ionization mass spectrometer. From the measured signals, the cluster evaporation rates were retrieved from model calculations (Jen et al., 2016a). The main differences to the CLOUD study lie within the much shorter reaction time (20 s vs. steady state in CLOUD) and in the much wider range of base to acid ratios used by Jen et al. (2016a, 2016b). This allowed them to retrieve even relatively slow evaporation rates for the sulfuric acid-dimethylamine clusters. The measured cluster/particle concentrations increased with increasing base to acid ratio, eventually approaching a plateau at a dimethylamine to acid ratio of ~1. Therefore, the high dimethylamine to acid ratio used in the CLOUD7 experiment (~ 100) can probably explain why our NPF rates are compatible with collision-controlled nucleation.

However, this was further tested by incorporating the evaporation rates from Jen et al. (2016a) in our model. For this purpose, the model described in section 2.4 was modified in a way that allows retrieving the cluster concentrations of the monomer, dimer, trimer and tetramer as a function of their dimethylamine content (see Appendix A). The abbreviation $A_xB_y$ denotes the concentration of a cluster containing $x$ sulfuric acid ($x = 1$ for the monomer) and $y$ base (dimethylamine) molecules. It is assumed that $x \geq y$ for all clusters, i.e., the number of bases is always smaller or equal to the number of acid molecules. The reported cluster concentrations (Fig. 3) refer to the number of acid molecules in the cluster, i.e., $N_1 = A_1 + A_1B_1$, $N_2 = A_2B_1 + A_2B_2$ and $N_3 = A_3B_1 + A_3B_2 + A_3B_3$.

The evaporation rates considered are $k_{e,A1B1} = 0.1$ s$^{-1}$, $k_{e,A3B1} = 1$ s$^{-1}$, $k_{e,A3B2} = 1$ s$^{-1}$ (Jen et al., 2016a). Jen et al. (2016a) suggested that the formation of stable tetramers requires at least two base molecules. In this case the evaporation rate of $k_{e,A4B1}$ is infinity. In the model, this was solved by not taking into account the formation of clusters $A_4B_1$ (from $A_3B_1$ and $A_1$) at all. Further details about the modeling involving evaporation rates can be found in Appendix A and in Table 1, which gives a summary over the different model studies.

**3. RESULTS**

**3.1 Comparison between Almeida et al. (2013) and SMPS derived NPF rates**

Using the model described in section 2.4, a comparison between the previously published NPF
rates from Almeida et al. (2013) and the modeled rates was performed. Almeida et al. (2013)
derived NPF rates for a particle mobility diameter of 1.7 nm. Using a density of 1470 kg m$^{-3}$
and a molecular weight of 0.143 kg mol$^{-1}$, it can be calculated that a spherical cluster containing
nine monomers (nonamer) has a geometric diameter of ~1.4 nm, i.e., a mobility diameter of 1.7
nm (Ku and Fernandez de la Mora, 2009, see also Appendix A); therefore, the modeled nonamer
formation rates were used for the comparison.
Figure 1 shows the modeled formation rates at 1.7 nm and the Almeida et al. (2013) data as
a function of the sulfuric acid concentration (which is equivalent to the monomer concentration
in the model, see section 2.4, since it is assumed that all sulfuric acid is bound to DMA). It can
be seen that the modeled NPF rates are significantly higher. This indicates that the previously
published formation rates underestimate the true formation rates if sulfuric acid-dimethylamine
nucleation is indeed proceeding at the collision-limit. Previously published results indicated
that this is the case (Kürten et al., 2014; Lehtipalo et al., 2016); however, we will provide further
evidence that this assumption accurately describes the experiments in the present study and
provide an explanation why Almeida et al. (2013) underestimated the formation rates.
It should be noted that the displayed experimental $J_{1.7nm}$ values (open red triangles in Fig. 1)
are identical to the values from Almeida et al. (2013), while the sulfuric acid concentration has
been corrected. In Almeida et al. (2013) data were shown from CLOUD4 (spring 2011) and
CLOUD7 (fall 2012). For consistency, the sulfuric acid concentrations from the chemical
ionization mass spectrometer (Kürten et al., 2011) were used, as the CI-APi-TOF was not
available during CLOUD4. Especially during CLOUD7, the chemical ionization mass
spectrometer (CIMS) showed relatively high sulfuric acid concentrations even when no sulfuric
acid was produced from the UV light system inside the CLOUD chamber; no correction was
applied for this effect in Almeida et al. (2013). However, taking into account a subtraction of
this instrumental background (reaching sometimes values above $1\times10^6$ cm$^{-3}$) leads to a
shallower slope for $J_{1.7nm}$ vs. sulfuric acid and brings the corrected CIMS values in a good
agreement with the sulfuric acid measured by the CI-APi-TOF. In the present study, the data
from the CI-APi-TOF were used. The slope for $J_{1.7nm}$ vs. sulfuric acid now yields a value of
close to 2, while the previously reported value was ~3.7 (Almeida et al., 2013). The higher
value resulted from the bias in the sulfuric acid concentration and the consideration of data
points at low sulfuric acid concentration, where new particle formation is significantly affected
by losses to the chamber walls, which tends to bias the slope towards higher values (Ehrhart
and Curtius, 2013).

**3.2 Comparison between NPF rates from the kinetic model and SMPS measurements**

The formation rates in Almeida et al. (2013) were calculated from measured particle number
densities with a condensation particle counter that has a lower cut-off diameter of ~3 nm. The
derivation of particle formation rates at 1.7 nm therefore required an extrapolation to the smaller
diameter (Kerminen and Kulmala, 2002). With the available model, we are now, in principle,
able to calculate NPF rates for any particle dimeter and compare the result to directly measured
rates. This was done for the SMPS size channel corresponding to a mobility diameter of 4.3 nm
($J_{4.3nm}$) with the method described in section 2.2. Using the SMPS data has the advantage that
the size-dependent loss rates can be accurately taken into account, which is not possible when
only the total (non size-resolved) concentration from a condensation particle counter is
available. On the other hand, the smallest SMPS size channels need to be corrected by large
factors to account for losses and charging probability (section 2.1), which introduces
uncertainty.
The result for $J_{4.3nm}$ is shown in Figure 1 together with the modeled particle formation rates
for the same diameter. The agreement between modeled and measured NPF rates is very good
indicating that the collision-controlled model accurately describes 4.3 nm particle production
rates for these experiments. This is further evidence that particles are formed at the collision-
limit. However, it is also an indication that the Almeida et al. (2013) data underestimate the
NPF rates, which is further discussed in the following section.

**3.3 Reconstruction model results**

Recently, a new method was introduced, which allows the extrapolation of NPF rates
determined at a larger size ($d_{p2}$) to a smaller diameter ($d_{p1}$). The advantage of that method is
that the effect of cluster-cluster collisions (self-coagulation) can be accurately taken into
account (Kürten et al., 2015a). So far, the method has not been tested for measured particle size
distributions. However, the effect of cluster-cluster collisions should be largest in the case of
collision-controlled nucleation since it results in the highest possible cluster (particle)
concentrations for a given production rate of nucleating molecules. Therefore, the current data
set is ideal for testing the new method. It requires the measured growth rate as an input
parameter (equation (4)); this growth rate was derived from fitting a linear curve to the mode
diameter determined from the SMPS size distribution (Hirsikko et al., 2005). It was then used
as a constant (i.e., it was assumed that it is independent of size) for the full reconstruction of
the size distribution, in order to obtain a formation rate at 1.7 nm. The growth rate could only
be accurately determined for experiments with relatively high sulfuric acid concentration
(above ~$5\times10^6$ cm$^{-3}$); therefore, the reconstruction method was only tested for these conditions
(Figure 1). The comparison with the modeled formation rates at the same size (1.7 nm) shows
that the reconstruction method yields quite accurate results, highlighting the importance of
cluster-cluster collisions in this chemical system. This explains why the Almeida et al. (2013)
data strongly underestimate the particle formation rates.
While the reconstruction method gives good results in the present study, it needs to be
mentioned that the errors for this method can become quite large. Small inaccuracies in the
growth rate, can be blown up to very large uncertainties due to the non-linear nature of the
method. This can be seen for some of the data points with large error bars in the positive
direction. The errors are calculated by repeating the reconstruction with growth rates $GR \pm \mathrm{d}GR$,
where $\mathrm{d}GR$ ($\pm$ 20%) is the error from the fitted growth rate. Therefore, the accuracy of the
method strongly depends on good growth rate measurements, and relies on the assumption that
the growth rate does not change as a function of size. This seems to be a reasonable
approximation for collision-controlled nucleation under the present conditions (Kürten et al.,
2015a), but it could be different in other chemical systems.
The higher formation rates are also consistent with calculations from the ACDC
(Atmospheric Cluster Dynamics Code) model (McGrath et al., 2012) that were previously
published in Almeida et al. (2013). Figure 1 shows the rates calculated by the ACDC model
(black lines). It should be noted that these values refer to a mobility diameter of 1.2 to 1.4 nm
and therefore, somewhat higher rates are expected due to the smaller diameter compared to
$J_{1.7nm}$. However, the agreement between the measured and predicted rates from ACDC are now
in much better agreement than before.
Hanson et al. (2017) recently reported an expression for the calculation of particle formation
rates as a function of the sulfuric acid concentration, dimethylamine concentration and
temperature. According to their formula the formation rate of tetramers (mobility diameter of
~1.4 nm, see Appendix A) follows the expression

$$J_{1.4nm} = exp\left(-129 + \frac{16200\,K}{T}\right) \cdot \left(\frac{N_1}{cm^{-3}}\right)^3 \cdot \left(\frac{DMA}{cm^{-3}}\right)^{1.5}. \qquad (10)$$

The formation rates $J_{1.4nm}$ are shown in Fig. 1 (green line) for a DMA mixing ratio of 40 pptv
($1\times10^9$ cm$^{-3}$) and a temperature of 278 K. At the first glance, the agreement between the
experimental CLOUD data and the ACDC simulation is remarkably good. However, one should
note that Hanson et al. (2017) recommended to use their equation only for DMA between 2
pptv ($5\times10^7$ cm$^{-3}$) and 16 pptv ($4\times10^8$ cm$^{-3}$) if sulfuric acid is present between $1\times10^6$ cm$^{-3}$ and
$2\times10^7$ cm$^{-3}$. Using the equation in this range avoids that the formation rates can exceed the
kinetic limit. When using larger concentrations, the kinetic limit is eventually exceeded due to
the power dependency of 3 regarding sulfuric acid and the 1.5 power dependency for DMA.
Further comparison between equation (10) and the results from the present study are shown in
Fig. 3 (lower panel).

**3.4 Size distribution comparison between model and SMPS**

Further comparison between modeled and measured data was performed for one experimental
run (CLOUD7 run 1036.01) in which the particles were grown to sizes beyond 20 nm.
Therefore, the time-dependent cluster/particle concentrations were modeled for a monomer
production rate of $2.9\times10^5$ cm$^{-3}$ s$^{-1}$, which results in a steady-state monomer concentration of
$1.07\times10^7$ cm$^{-3}$ for the model; this is the same as the measured sulfuric acid concentration. The
measured and modeled size distributions are shown in Fig. 2 (panels a, b and c) at four different
times, i.e., at 1h, 2h, 4h and 6h after the start of the experiment. Given that there is no free
parameter used in the model, the agreement between the base case simulation and the
measurement is very good (Fig. 2a). For the earliest time shown (1h) the modeled
concentrations overestimate the measured concentrations by up to 30%, whereas for the later
times ($\geq$ 4h) the model underestimates the measured concentrations by up to 30%. It is unclear
whether these discrepancies are due to SMPS measurement uncertainties, or if the model does
not include or accurately describe all the relevant processes. If, for example, the SMPS would
underestimate the concentrations of the smaller particles (< ca. 15 nm) and overestimate those
of the larger particles, the observed difference between modeled and measured concentrations
could also be explained.
A comparison between measured and modeled aerosol volume concentrations is shown
in Fig. 2d. In order to enable direct comparison, the modeled size distribution was integrated
starting at 4.3 nm since the SMPS did not capture smaller particles. In the beginning of the
experiment the modeled aerosol volume is up to ~40% larger than the measured one, but,
towards the end of the experiment (ca. 4h after its start), the volumes agree quite well. Possibly
this is because the overestimated modeled particle number density at small diameters is
compensated by the underestimated particle concentration in the larger size range (see Fig. 2a).
This trend leads eventually to a slight underestimation of the aerosol volume by the model.
If one assumes that the SMPS is not responsible for the slight disagreement, then the
following conclusions can be drawn regarding the accuracy of the model. The particle growth
rate is well represented by the model given the good agreement between the positions of the
local maxima in the size distribution and the intersections between the size distributions and
the $x$-axis. This good agreement between measured and modeled growth rates has already been
demonstrated in Lehtipalo et al. (2016) for a particle diameter of 2 nm. The results shown here
indicate that no significant condensation of other trace gases contribute to the growth of
particles because, in this case, the measured particle size distributions would be shifted towards
larger diameters compared to the model.
The good agreement between model and measurement is also a confirmation of the effect of
van der Waals forces, when a Hamaker constant of $6.4 \times 10^{-20}$ J is used, a value that has been
demonstrated previously to represent particle size distribution dynamics correctly (McMurry,
1980; Chan and Mozurkewich, 2001; Kürten et al., 2014; Lehtipalo et al., 2016). Regarding the
underestimation of the modeled size distribution for diameters $\gtrsim$15 nm, one explanation could
be that the size-dependent particle loss rates in the CLOUD chamber are weaker than assumed
($k_w \sim D^{0.5}$; see equation (2)). A weaker size dependence would lead to higher predicted particle
concentrations at larger sizes (Park et al., 2001). However, no evidence was found from the
existing CLOUD data that this is the case. Dedicated wall loss experiments could be performed
in the future to investigate this hypothesis further.
In order to test the model sensitivity to certain variations quantitatively further simulations
were performed (Fig. 2b and Fig. 2c). A variation of the steady-state sulfuric acid monomer
concentration by ±20% was achieved by using different monomer production rates for the high
sulfuric acid case ($P_1 = 4.17 \times 10^5$ cm$^{-3}$ s$^{-1}$) and for the low sulfuric acid case ($P_1 = 2.01 \times 10^5$
cm$^{-3}$ s$^{-1}$, Fig. 2b). This rather small variation leads to significant mismatches between the
modeled and measured size distributions that is also found for the aerosol volumes (Fig. 2d).
Two further scenarios were tested with the model. First, the enhancement due to van der
Waals forces were turned off. This scenario results in significantly slower growth rates and the
modeled size distributions do not match the measured ones at all anymore (Fig. 2c); the same
is found when comparing modeled and measured aerosol volumes (Fig. 2d). Second, the aerosol
density and the molecular weight of the condensing "monomer" were changed. In the base-case
simulations (Fig. 2a), the density of dimethylaminium-bisulfate is 1470 kg m$^{-3}$ and the
molecular weight is 0.143 kg mol$^{-1}$ because a one to one ratio between DMA and sulfuric acid
is assumed. Since full neutralization of sulfuric acid by DMA would require a 2:1-ratio between
base and acid, collision-controlled nucleation of $(H_2SO_4)((CH_3)_2NH)_2$ "monomers" instead of
$(H_2SO_4)((CH_3)_2NH)$ was tested. Therefore, the density was decreased by 6% to account for the
density change between dimethylaminium-bisulfate and dimethylaminium-sulfate (see Qiu and
Zhang, 2011) and the molecular weight was set to 0.188 kg mol$^{-1}$. As expected, the particle
growth is now slightly faster due to the additional volume added by the further DMA molecules

(Fig. 2c). However, the changes are rather small and the modeled size distributions move a little further away from the measurements compared to the base case scenario (Fig. 2a).

Comparison between modeled and measured size distributions yielded similar results for other experiments from CLOUD7. However, the experiment shown in Fig. 2 was carried out over a relatively long time (6 h) at high sulfuric acid concentrations. Therefore, the particles could grow to large diameters and the comparison between model and experiment covers a wide size range.

## 3.5 Sensitivity of cluster concentrations and NPF rates regarding DMA

The data presented in the previous sections provide evidence that the new particle formation in the sulfuric acid-dimethylamine system during CLOUD7 proceeds at rates that are consistent with collision-controlled nucleation, in agreement with results for this data set obtained using different approaches (Kürten et al., 2014; Lehtipalo et al., 2016). In this section, we compare whether for CLOUD conditions the collision-controlled assumption is consistent with the Jen et al. (2016a) results that showed that some clusters evaporate at the rates given in section 2.5 and Table. 1.

For the following discussion, both versions of the nucleation and growth model (section 2.4 and section 2.5) were used. Figure 3 shows a comparison between calculated cluster (dimer, trimer, tetramer and pentamer) concentrations using collision-controlled nucleation (section 2.4) and the model described in section 2.5. When a DMA mixing ratio of 40 pptv ($1 \times 10^9$ cm$^{-3}$) is used (this was the average mixing ratio of DMA during the CLOUD7 experiments), there is almost no difference between the two scenarios. This indicates that, under the CLOUD7 conditions, new particle formation proceeded at almost the same rates that result for collision-controlled nucleation. Nevertheless, this does not imply that all cluster evaporation rates are zero. The conditions are only such that, due to the high DMA mixing ratio, most of the clusters (including the monomer) probably contain as many DMA molecules as sulfuric acid molecules; this results in very stable cluster configurations (Ortega et al., 2012). When DMA mixing ratios are low, most sulfuric acid clusters contain, however, only a small number of DMA molecules. As these clusters can evaporate more rapidly, the overall formation rate is slowed down (Ortega et al., 2012; Hanson et al., 2017). For low base to acid ratios, it can therefore matter whether a cluster is stabilized by a dimethylamine, a diamine (Jen et al., 2016) or by both an amine and an ammonia molecule (Glasoe et al., 2015). This can explain the more efficient NPF due to diamines or the synergistic effects involving amines and ammonia at low base to acid ratios. At high base to acid ratios, the differences in the effective evaporation rates become small (Jen et al., 2016b).

The effect of the dimethylamine concentration on the cluster concentrations and on the particle formation rate was further investigated. The lower panel of Fig. 3 shows that the cluster concentrations and the NPF rate at 1.7 nm decrease with decreasing DMA levels. The figure shows the concentrations and the NPF rate normalized by the results for the collision-limit. The NPF rate drops by about a factor of three when DMA is reduced to $2.5 \times 10^7$ cm$^{-3}$ (~ 1 pptv). Below that level, the reduction in $J$ and in the trimer, tetramer, and pentamer concentrations is approximately linear with DMA. The dimer is less affected since, in the model, its evaporation rates are set to zero while the evaporating trimers contribute to the dimer concentration. From

this perspective, very high particle formation rates should be observed even at DMA mixing
ratios around 1 pptv ($2.5 \times 10^7$ cm$^{-3}$), which should be almost indistinguishable from rates
calculated for collision-controlled nucleation. Possibilities why such high rates have so far not
been observed are discussed in section 4.
For a comparison, the expected formation rates from equation (10) are shown in Fig. 3, lower
panel, by the grey line. The values were scaled similar to the simulated data by setting the value
for 40 pptv ($1 \times 10^9$ cm$^{-3}$) to 1. Although this DMA mixing ratio is outside the range for which
the Hanson et al. (2017) formulation is recommended for (between $5 \times 10^7$ cm$^{-3}$ and $4 \times 10^8$ cm$^{-3}$),
from Fig. 1 it can be concluded that both, the Hanson et al. (2017) equation and the kinetic
model agree quite well at this DMA mixing ratio. The slope of $J$ vs. DMA seems to be, however,
different in the relevant range of DMA ($5 \times 10^7$ cm$^{-3}$ and $4 \times 10^8$ cm$^{-3}$). This is due to the fact,
that the model predicts a steep slope (close to the value of 1.5 in equation (10)) only for much
lower DMA ($< 2.5 \times 10^6$ cm$^{-3}$), for higher DMA the slope flattens out and reaches eventually a
plateau, when the value for collision-controlled nucleation is approached. This flatting of the
curve is not reflected in the simple formulation from Hanson et al. (2017). However, in contrast
to the three constant evaporation rates used in our modeling approach, Hanson et al. (2017)
used a more sophisticated nucleation scheme involving many different evaporation rates, not
only regarding sulfuric acid but also for dimethylamine. This more complex scheme was,
however, not implemented in our model.
Further experiments are required to derive accurate values for evaporation rates in the
sulfuric acid-dimethylamine system; these experiments should especially target DMA
concentrations with low base to acid ratios ($< 10$).
**4. DISCUSSION**
This study confirms the results derived in previous studies that new particle formation in the
sulfuric acid-dimethylamine-water system can proceed at or close to the collision-controlled
limit (Kürten et al., 2014; Lehtipalo et al., 2016). This is the case for sulfuric acid concentrations
between $1 \times 10^6$ and $3 \times 10^7$ cm$^{-3}$ and dimethylamine mixing ratios around 40 pptv ($1 \times 10^9$ cm$^{-3}$)
at 278 K and 38% RH. For these conditions particle formation rates and size distributions can
be reproduced with high accuracy by an aerosol model that assumes that particle growth is
exclusively due the irreversible addition of $H_2SO_4 \cdot (CH_3)_2NH$ "monomers" and coagulation.
Even when evaporation rates for the less stable clusters are introduced in the model (Jen et al.,
2016a) the resulting particle formation rates are effectively indistinguishable from the kinetic
model results for CLOUD7 conditions (i.e., at the high dimethylamine to acid ratio of ~100).
The fact that the measured particle size distribution can be reproduced with good accuracy
shows that neither water nor other species contribute significantly to particle growth during
these CLOUD chamber experiments. Water could play a role at higher relative humidities,
although quantum chemical calculations suggest that it plays only a minor role in NPF for the
system of sulfuric acid and dimethylamine (Olenius et al., 2017); this contrasts the sulfuric acid-
water system (see e.g. Zollner et al. 2012; Duplissy et al., 2016; Yu et al., 2017). In addition, it
is not exactly known how temperature influences the cluster evaporation rates (Hanson et al.,
2017). The evaporation rates from Jen et al. (2016a) were derived at temperatures close to 300
K; therefore the simulation of nucleation in the CLOUD chamber (278 K) using the Jen et al.
(2016a) rate parameters is likely to overestimate the effect of cluster evaporation.
It is not yet clear what exact base to acid ratio the particles have for a given diameter. The
clusters and small particles (< ~2 nm) seem to grow by maintaining a 1:1 ratio between base
and acid, which follows from measurements using mass spectrometers (Almeida et al., 2013;
Kürten et al., 2014; Bianchi et al., 2014). The larger particles could eventually reach a 2:1 ratio
between base and acid, especially at the DMA mixing ratios relevant for this study (Ahlm, et
al., 2016). However, even when a 2:1 ratio is assumed in the model (Fig. 2c) the expected size
distributions would not change significantly compared with the base-case scenario (1:1 ratio).
Therefore, it is not possible from our comparisons to find out if and at what diameter a transition
from 1:1 to 2:1 base to acid ratio takes place.
The question of why sulfuric acid-amine nucleation is rarely observed in the atmosphere is
still open. Jen et al. (2016a) reported that clusters that contain equal numbers of dimethylamine
and sulfuric acid molecules are ionized at reduced efficiencies than more acidic clusters with
the commonly used $NO_3^-(HNO_3)_{0-2}$ reagent ions. Still, Kürten et al. (2014) observed high
concentrations for large clusters containing acid and base at an average ratio of 1:1. A reduced
detection efficiency was also reported but the reduced sensitivity (in relation to the monomer)
was, e.g., only a factor of 3 for the trimer containing DMA. Using the model results from section
3.5 the expected trimer concentration at $5\times10^6$ cm$^{-3}$ of sulfuric acid and 1 pptv ($2.5\times10^7$ cm$^{-3}$)
of DMA should be ~$1\times10^5$ cm$^{-3}$. Even when the detection efficiency for the trimer was reduced
by a factor of 3, such a concentration should still be well above the detection limit of a CI-APi-
TOF. However, no sulfuric acid trimers could be detected in a field study where amines were
present at levels above 1 pptv ($2.5\times10^7$ cm$^{-3}$, Kürten et al., 2016b). It is, therefore, possible that
any amines present were not suitable for nucleation. Therefore, application of methods capable
of amine speciation should be applied more widely in atmospheric measurements (Place et al.,
2017).

Several CLOUD papers reported particle formation rates for a diameter of 1.7 nm. Some of
these published formation rates were derived from direct measurements using particle counters
with cut-off diameters close to 1.7 nm (Riccobono et al., 2014; Duplissy et al., 2016), while
other reported NPF rates were derived from process models describing the nucleation process
in the CLOUD chamber (Kirkby et al., 2011; Kirkby et al., 2016). Therefore, no extrapolation
of the NPF rates from a larger threshold diameter was performed, which could have led to an
underestimation due to missing self-coagulation. Besides Almeida et al. (2013), the data set
reported by Dunne et al. (2016) and Kürten et al. (2016a) did make use of the NPF rate
extrapolation method from 3.2 to 1.7 nm without taking into account the effect of self-
coagulation. However, the reported formation rates are, in almost all cases, considerably slower
than those for the collision-controlled limit at a given sulfuric acid concentration since no
dimethylamine was present in the CLOUD chamber (Dunne et al., 2016; Kürten et al., 2016a).
The chemical system in these studies was the binary system, ($H_2SO_4$ and $H_2O$) and the ternary
system involving ammonia. The conditions only approached the collision-controlled limit at
the lowest temperature (210 K) when the highest ammonia mixing ratio of ~6 pptv ($1.5\times10^8$
cm$^{-3}$) was investigated (Kürten et al., 2015b). However, even under these conditions, the
reported rates are only about a factor of 2 slower than the collision-controlled limit (Kürten et
al., 2016a). This is probably related to the low acid concentrations ($\leq 3\times10^6$ cm$^{-3}$) in these

experiments, where the self-coagulation effect is not as strong as at higher acid concentration (see Fig. 1) when wall loss and dilution lead to decreased cluster concentrations relative to the monomer. This indicates that previously published CLOUD results, other than the Almeida et al. (2013) data, are most likely not significantly affected.

McMurry and Li (2017) have recently investigated the effect of the wall loss and dilution rate on new particle formation with their numerical model, which uses dimensionless parameters. In order to allow for a comparison between McMurry and Li (2017) and the present study, information on the dimensionless parameters $W$ (describing wall loss) and $M$ (describing dilution) is provided (see McMurry and Li, 2017, for the exact definitions). These parameters range from 0.04 to 0.7 ($W$) and $2\times10^{-3}$ to $4\times10^{-2}$ ($M$) for the experiments shown in this study (Fig. 1). The monomer production rate ($P_1$) ranges from $7\times10^3$ to $2\times10^6$ cm$^{-3}$ s$^{-1}$.

## 5. SUMMARY AND CONCLUSIONS

New particle formation rates from CLOUD chamber measurements for the sulfuric acid-dimethylamine-water system were re-analyzed. It was found that the previously published rates by Almeida et al. (2013) underestimate the NPF rates by up to a factor of ~50 at high acid concentrations (~$1\times10^7$ cm$^{-3}$). The reason for this underestimation is the effect of self-coagulation that contributes efficiently to the loss of small particles in the size range relevant for the data analysis (between 1.7 and 3.2 nm). The previously used method for extrapolating the NPF rates from 3.2 nm to 1.7 nm did not include this effect and therefore the correction factors were too small. Using an advanced reconstruction method that accounts for the effect of self-coagulation yields much higher NPF rates (Kürten et al., 2015a). These corrected NPF rates are in good agreement with rates calculated from an aerosol model assuming collision-controlled nucleation and with measured NPF rates from SMPS data. Furthermore, the model can reproduce the measured size distribution with good accuracy up to ~30 nm.

Extending the aerosol model by including evaporation rates for some clusters (see Jen et al., 2016a) still yields good agreement between modeled and measured CLOUD NPF rates and cluster concentrations. This indicates that the data for sulfuric acid-dimethylamine from the flow tube study by Jen et al. (2016a) and from CLOUD (Kürten et al., 2014) are consistent for the high base to acid ratio relevant for this study (dimethylamine to sulfuric acid monomer ratio of ~100).

The above findings raise some further conclusions and questions. These are in part related to the rare detection of sulfuric acid-amine nucleation in the atmosphere. Only one study has so far reported sulfuric acid-amine nucleation (Zhao et al., 2011). The nucleation of sulfuric acid-amines could occur, however, more often than currently thought.

–   It is unclear to what extent previously published atmospheric NPF rates are affected by incomplete $J$ extrapolations. Some $J$ measurements were made at diameters close to 3 nm and extrapolated to a smaller size. If self-coagulation were important, the formation rates at the small sizes could be significantly underestimated, and, therefore, in reality be much closer to rates consistent with collision-controlled nucleation than previously thought. In such a case, DMA (or other equally effective amines) could have been responsible for

nucleation as they are among the most potent nucleation precursors (in combination with
sulfuric acid). To avoid such ambiguities, the NPF rates should, in the future, be directly
measured at small diameters whenever possible.

–   Better gas-phase amine (base) measurements are needed. Detection limits need to reach
mixing ratios even below 0.1 pptv ($2.5 \times 10^6$ cm$^{-3}$); ideally the methods should also be
capable of speciating the amines (discriminate e.g. dimethylamine from ethylamine, which
have the same mass when measured by mass spectrometry but probably behave differently
in terms of their contribution to NPF). High time resolution (several minutes or better) for
the amine measurements during nucleation events is also important. This can show, whether
amines can be significantly depleted during NPF. As amines are not produced in the gas
phase (unlike sulfuric acid), their clustering with sulfuric acid monomers and small sulfuric
acid clusters/particles very likely can lead to a significant reduction in the amine mixing
ratios (Kürten et al., 2016b). This would indicate that new particle formation involving
amines in the atmosphere could be self-limiting, i.e., after an initial burst of particles, new
particle formation could be slowed down soon after when amine mixing ratios decrease.

–   It is not clear why no clusters containing three or more sulfuric acid molecules are frequently
observed during atmospheric new particle formation when amines are expected to be
present. This could be due to incorrect assumptions about the amine concentrations, the
amine identities, or a reduced detection efficiency of chemical ionization mass
spectrometers (Jen et al., 2016a). The potential formation of complex multi-species clusters
(containing sulfuric acid, amines, ammonia and oxidized organics) in the atmosphere could
distribute the clusters over many different identities and therefore result in concentrations
too low to be detected by the current instrumentation for the individual species.


The overall contribution of amines to atmospheric nucleation can only be quantified after these
issues are understood. Besides further atmospheric measurements, controlled laboratory
measurements are necessary. Of special interest are the temperature dependent evaporation
rates of the relevant sulfuric-acid amine (and diamine) clusters.

**Appendix A:**

**Model that includes selected evaporation rates**

The kinetic model described in section 2.4 was expanded in a way that allows calculating the concentrations of the monomer, dimer, trimer and tetramer as a function of their dimethylamine content. Here, $A_xB_y$ denotes the concentration of a cluster containing $x$ sulfuric acid ($x = 1$ for the monomer) and $y$ base ($y = 1$ for dimethylamine monomer) molecules; $x \geq y$ for all clusters, i.e., the number of bases is always smaller or equal to the number of acid molecules. When the total monomer concentration ($N_1$) is fixed, i.e., $A_1 = N_1 - A_1B_1$ at each time step, then the following equations result, i.e., for the $A_1B_1$ cluster

$$\frac{dA_1B_1}{dt} = K_{1,1} \cdot B_1 \cdot A_1 - \left(k_{1,w} + k_{dil} + k_{e,A_1B_1} + \sum_{j=1}^{N_{max}} K_{1,j} \cdot N_j\right) \cdot A_1B_1, \tag{A1}$$

for the two different identities of the sulfuric acid dimer

$$\frac{dA_2B_1}{dt} = \left(K_{1,1} \cdot A_1 \cdot A_1B_1 + k_{e,A_3B_1} \cdot A_3B_1\right) - \left(k_{w,2} + k_{dil} + K_{1,2} \cdot B_1 + \sum_{j=1}^{N} K_{j,2} \cdot N_j\right) \cdot$$
$$A_2B_1, \tag{A2}$$

$$\frac{dA_2B_2}{dt} = \left(0.5 \cdot K_{1,1} \cdot A_1B_1 \cdot A_1B_1 + K_{1,2} \cdot B_1 \cdot A_2B_1 + k_{e,A_3B_2} \cdot A_3B_2\right) - \left(k_{w,2} + k_{dil} + \sum_{j=1}^{N} K_{j,2} \cdot N_j\right) \cdot A_2B_2, \tag{A3}$$

and for the three different identities of the sulfuric acid trimer

$$\frac{dA_3B_1}{dt} = \left(K_{1,2} \cdot A_1 \cdot A_2B_1\right) - \left(k_{w,3} + k_{dil} + k_{e,A_3B_1} + K_{1,3} \cdot B_1 + \sum_{j=1}^{N} K_{j,3} \cdot N_j - K_{1,3} \cdot A_1\right) \cdot$$
$$A_3B_1, \tag{A4}$$

$$\frac{dA_3B_2}{dt} = \left(K_{1,2} \cdot A_1B_1 \cdot A_2B_1 + K_{1,2} \cdot A_1 \cdot A_2B_2 + K_{1,3} \cdot B_1 \cdot A_3B_1\right) - \left(k_{w,3} + k_{dil} + k_{e,A_3B_2} + K_{1,3} \cdot B_1 + \sum_{j=1}^{N} K_{j,3} \cdot N_j\right) \cdot A_3B_2, \tag{A5}$$

$$\frac{dA_3B_3}{dt} = \left(K_{1,2} \cdot A_1B_1 \cdot A_2B_2 + K_{1,3} \cdot B_1 \cdot A_3B_2\right) - \left(k_{w,3} + k_{dil} + \sum_{j=1}^{N} K_{j,3} \cdot N_j\right) \cdot A_3B_3. \tag{A6}$$

Since the formation of stable $A_4B_1$ clusters is not allowed (see Jen et al., 2016), the loss due to the $A_1$ and $A_3B_1$ collision is subtracted from the coagulation loss term in equation (A4).

Tetramers can be formed from trimers and dimers:

$$\frac{dN_4}{dt} = \left(K_{1,3} \cdot A_1B_1 \cdot A_3B_1 + K_{1,3} \cdot N_1 \cdot (A_3B_2 + A_3B_3) + 0.5 \cdot K_{2,2} \cdot N_2 \cdot N_2\right) - \left(k_{w,4} + k_{dil} + \sum_{j=1}^{N} K_{j,4} \cdot N_j\right) \cdot N_4. \tag{A7}$$

Note that the formation of $A_4B_1$ (from $A_3B_1$) is not included in the formation rate for tetramers (see also further below). The concentrations of larger clusters and particles are calculated with

the same method as described in section 2.4. The cluster concentrations reported in section 3.5
refer to the number of acid molecules in the cluster, i.e., $N_1 = A_1 + A_1B_1$, $N_2 = A_2B_1 + A_2B_2$ and
$N_3 = A_3B_1 + A_3B_2 + A_3B_3$.
The evaporation rates considered are $k_{e,A1B1} = 0.1$ s$^{-1}$, $k_{e,A3B1} = 1$ s$^{-1}$, $k_{e,A3B2} = 1$ s$^{-1}$ (Jen et al.,
2016a). Pure acid clusters are assumed to evaporate rapidly (at 278 K and higher) and are,
therefore, not considered (Hanson and Lovejoy, 2006). Jen et al. (2016a) suggested that the
formation of stable tetramers requires two base molecules. Therefore, this would indicate that
the evaporation rate $k_{e,A4B1}$ is infinity (or very fast), which is also shown by Hanson et al. (2017).
However, the $A_4B_1$ formation (and its evaporation) is not explicitly treated in equations (A4)
and (A7).
In summary, three different evaporation rates were included in this model version (equations
(A1) to (A7)), i.e., $k_{e,A1B1} = 0.1$ s$^{-1}$ (cluster $A_1B_1$), $k_{e,A3B1} = 1$ s$^{-1}$ (cluster $A_3B_1$) and $k_{e,A3B2} = 1$
s$^{-1}$ (cluster $A_3B_2$). All other evaporation rates were not explicitly included in the model, i.e.,
their rates were assumed to be zero (except for $A_4B_1$, which is assumed to be infinity). Table 1
gives an overview of the different model configurations used to generate the model data in the
figures.

**Calculation of particle mobility diameters**

The mobility diameter of a cluster containing $i$ sulfuric acid molecules (and $i$ DMA molecules)
can be calculated according to
$$d_{p,i} = \left(\frac{6 \cdot i \cdot M_w}{\pi \cdot N_A \cdot \rho}\right)^{1/3} + 0.3 \cdot 10^{-9} \, m. \qquad \text{(A8)}$$

$M_w$ is the molecular weight of the "monomer", i.e., 0.143 kg mol$^{-1}$, $\rho$ is the density of 1470 kg
m$^{-3}$ (see section 2.4) and $N_A$ is the Avogadro number, i.e., $6.022 \times 10^{23}$ mol$^{-1}$. The addition of
0.3 nm in equation (A8) is used to convert the geometric diameter (first term in equation (A8))
to a mobility diameter (Ku and Fernandez de la Mora, 2009).

**DATA AVAILABILITY**

Data used in this study can be obtained by sending an email to the corresponding author.

**ACKNOWLEDGEMENTS**

Funding from the German Federal Ministry of Education and Research (grant no. 01LK1222A) and the Marie Curie Initial Training Network "CLOUD-TRAIN" (grant no. 316662) is gratefully acknowledged. PHM's and CL's contributions to this work were supported by the US Department of Energy's Atmospheric System Research program, an Office of Science, Office of Biological and Environmental Research, under grant number DE-SC0011780. RCF acknowledges funding from the NSF Grants 1439551 and 1602086. MRP appreciates funding from the Academy of Finland (project no. 299574). KL thanks the European Union's Horizon 2020 research and innovation programme under the Marie Sklodowska-Curie grant agreement no. 656994 (nano-CAVa).

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

**Table 1.** Overview of the two different model versions used to generate the data in the figures.

| | kinetic model | model with evaporation rates |
|---|---|---|
| used for | Fig. 1, Fig. 2, Fig. 3 upper panel (black lines) | Fig. 3 upper panel (colored lines), Fig. 3 lower panel |
| described in | section 2.4 | section 2.5, Appendix A |
| evaporation rates | all zero | $k_{e,A1B1} = 0.1\ \text{s}^{-1}$ $k_{e,A3B1} = 1\ \text{s}^{-1}$ $k_{e,A3B2} = 1\ \text{s}^{-1}$ $(k_{e,A4B1} = \infty\ \text{s}^{-1})$ all others zero |

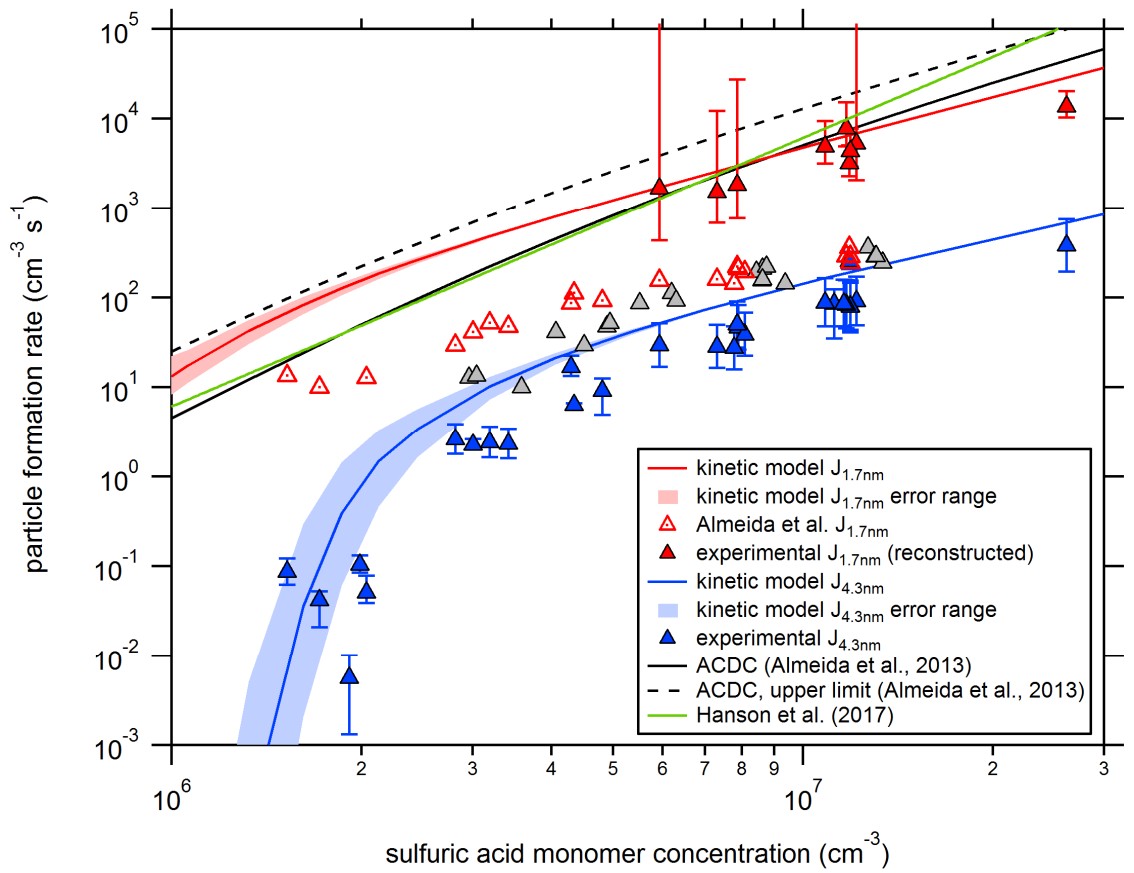

**Fig. 1.** Comparison between experimental and theoretical particle formation rates at different sizes (mainly at mobility diameters of 1.7 nm and 4.3 nm). The lines indicate calculated particle formation rates from the collision-controlled aerosol model described in section 2.4 for CLOUD chamber conditions. The shaded regions show the model uncertainties when using an error of ±20% for the wall loss coefficient ($C_w$, see equation (2)). The open red symbols show previously published CLOUD7 data for the sulfuric acid-dimethylamine-water system (Almeida et al., 2013), while the blue symbols show the rates derived from SMPS size distribution measurements (this study). The data shown by the closed red symbols were derived with the method introduced by Kürten et al. (2015a) by extrapolating the SMPS data starting at 4.3 nm. The black lines show the calculated formation rates from the ACDC model for a mobility diameter of 1.2 to 1.4 nm (Almeida et al., 2013). Equation (10) from Hanson et al. (2017) is used to generate the green line.

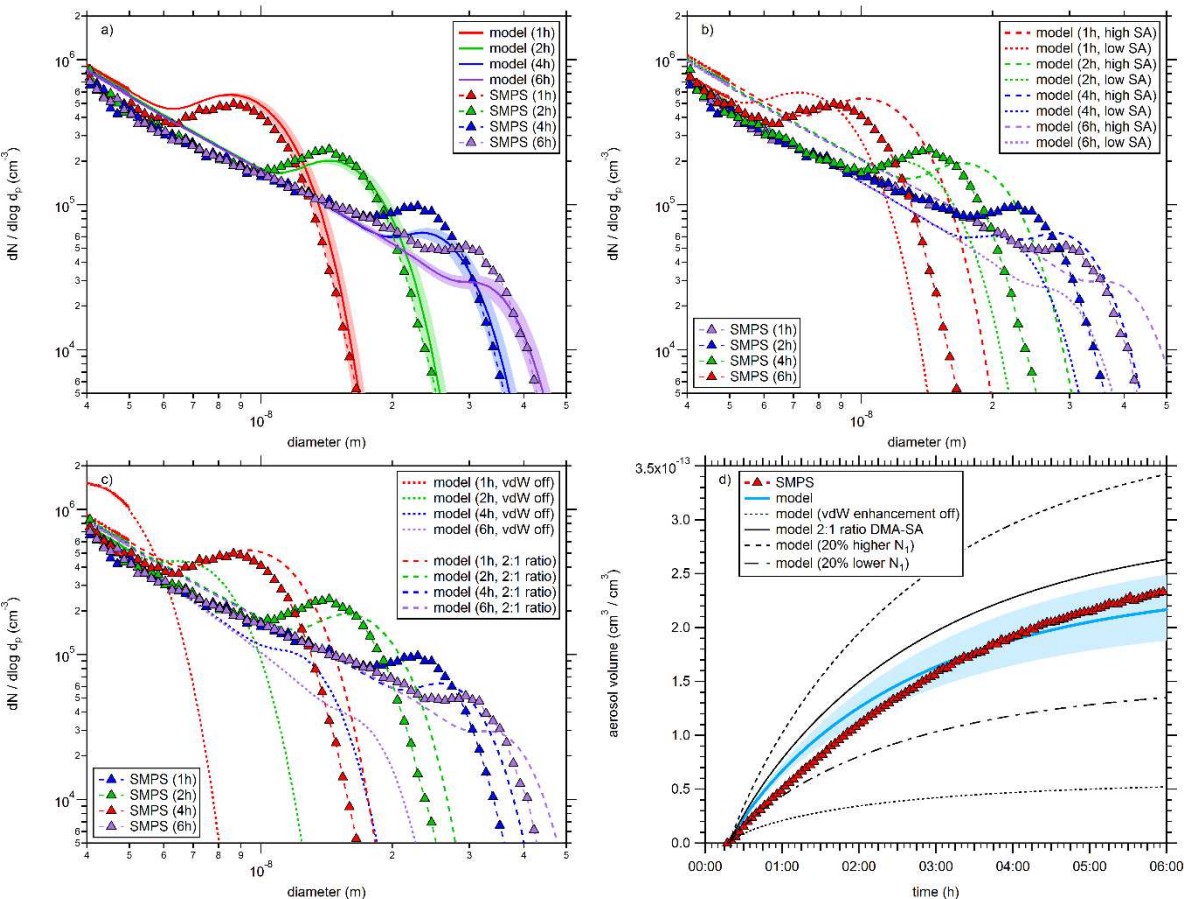

**Fig. 2.** Comparison between simulated and measured particle size distributions for one experiment (CLOUD7, run 1036.01). The comparison is shown for four different times (1h, 2h, 4h and 6h) after the start of the experiment (panels a, b and c). Panel d shows a comparison between modeled and measured aerosol volume as a function of time. The shaded regions in panel a show the model uncertainties when using an error of ±20% for the wall loss coefficient ($C_w$, see equation (2)). Panel b shows the change in the size distributions when the sulfuric acid monomer concentration is varied by ±20%. The effect of van der Waals forces on the size distribution is shown in panel c along with the assumption that particles grow by the addition of 2 DMA and 1 sulfuric acid molecule (2:1 ratio instead of 1:1 ratio). See text for further details.

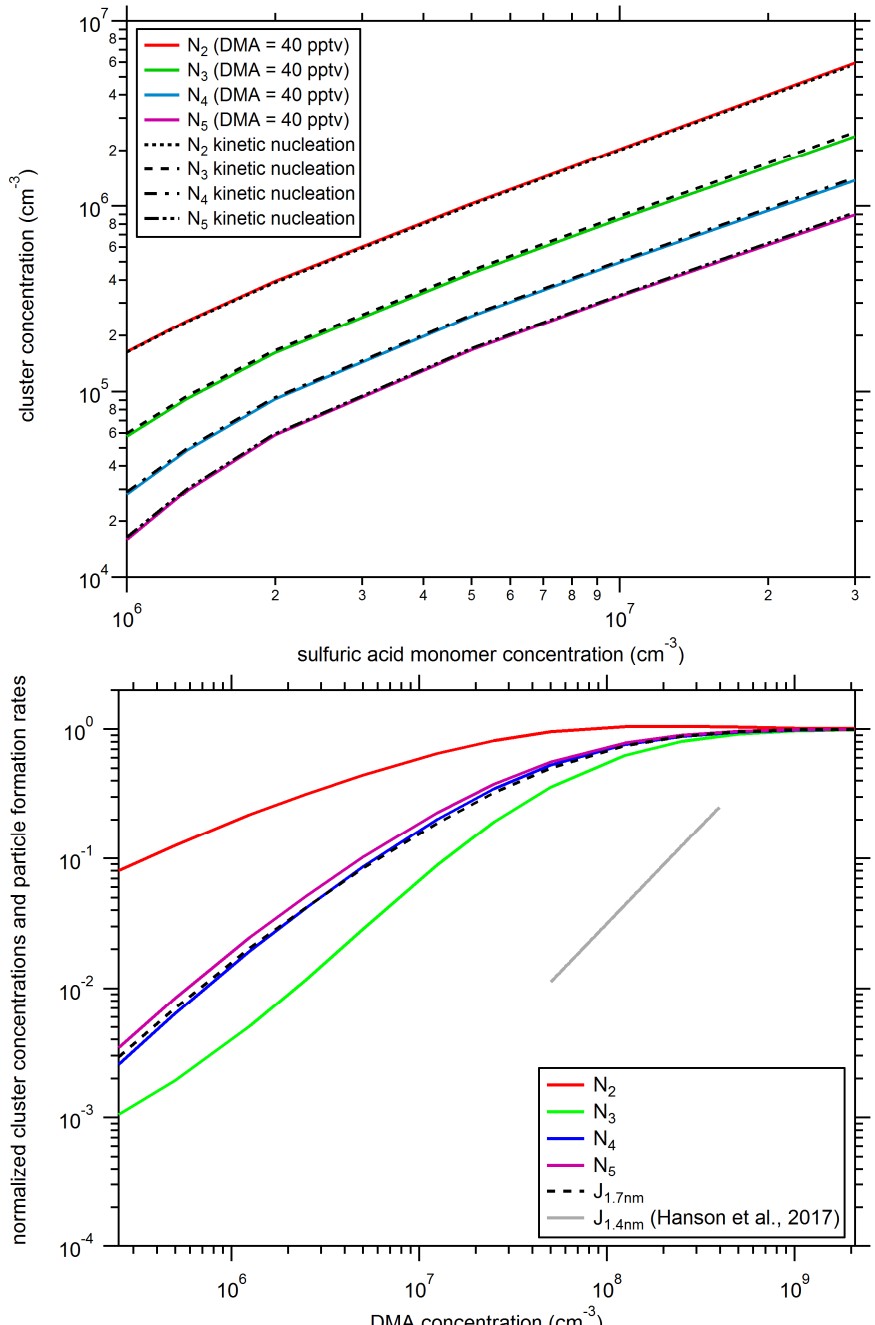

**Fig. 3.** Upper panel: Comparison of modeled cluster ($N_2$ = dimer, $N_3$ = trimer, $N_4$ = tetramer and $N_5$ = pentamer) concentrations using different scenarios. The dashed black lines use the collision-controlled nucleation scheme with all evaporation rates set to zero (section 2.4); while the colored solid lines are calculated based on the model from section 2.5 with a dimethylamine (DMA) mixing ratio of 40 pptv ($1 \times 10^9$ cm$^{-3}$), which was the average mixing ratio during the CLOUD7 campaign. Lower panel: Variation in modeled cluster concentration and $J_{1.7nm}$ as a function of the dimethylamine mixing ratio. The data were normalized to the values from the collision-controlled limit calculation (upper panel). For the calculations, a sulfuric acid monomer concentration of $N_1 = 5 \times 10^6$ cm$^{-3}$ was used. An expression from Hanson et al. (2017) to calculate NPF rates as a function of DMA is shown by the grey line. See text for further details.