# Peer review of "New particle formation in the sulfuric acid-dimethylamine-water system: Reevaluation of CLOUD chamber measurements and comparison to an aerosol nucleation and growth model"

_Atmospheric Chemistry and Physics, 2017_

## Referee Comment (RC1) · Anonymous Referee #2 · 10 Oct 2017

The study by Kürten et al. presents a revised analyses of their previous study on new particle formation of the sulfuric acid-dimethylamine-water system. The new results show that nucleation is even faster than previously thought and that the derived nucleation rates are in agreement with model simulations. Further, they show that even tiny mixing ratios of dimethylamine are sufficient to explain significant particle nucleation in the boundary layer.

The paper is well written and suitable for publication in ACP after some minor

revisions. My comments are listed below:

*General comments:*

Somewhere in the paper it should be stated what the typical concentrations/mixing ratios of dimethylamine and sulfuric acid in the boundary layer are. Are the concentrations that were used in your experiment realistic?

The difference between the Almeida et al. (2013) study and your study (same holds for the Jen et al. (2016) study) becomes not really clear from the manuscript as it is written now. If I understand your study correct your experiment is the same as the one by Almeida et al. (2013) but with an improved set-up which results in a better agreement between measurement and observations. This is of course very nice, but as you write it it sounds like "we are doing it right and Almeida et al. did it wrong" which is not correct. I am sure they did their best at the time they performed their study. Of course, with more time and more experience as well with improving knowledge previous studies can be repeated and improved. This should be more clearly and fair be discussed in the paper.

*Specific comments:*

P3, l36: add "mobility diameter" so that it reads ". . ....at a mobility diameter of 1.7 nm. . .. . ...".

P4, l47: The abundances of sulfuric acid and dimethylamine are given in different units (concentrations and mixing ratio, respectively). The same unit should be used for both consequently. However, I would prefer if the amounts of both species would be given as concentration and mixing ratio.

P5, l97 and l111: What the abbreviation CLOUD is standing for should be rather given in the introduction in l97 than in l111 of the method section.

P6, l144: The brackets around the reference are obsolete.

P6, l152: Same here, the brackets around the reference are obsolete.

P6, l169 andl172: Is the unit really correct? If yes, why is it s$^{-0.5}$?

P6, l300: Since you compare your results to Almeida et al. (2013) it would be good if you could write more about the Almeida et al. (2013) study. How were their nucleation rates derived. What are the differences between your experiment and their experiment? Were these experimentally derived or from model simulations?

P11, l369: Please give to the concentration the corresponding mixing ratio.

P12, L402ff: Concerning the discussion on the differences between model simulation and measurements. I would say that this part could be improved. Although I agree that the agreement is very good you should also admit that the nucleation rates from the model simulation are slightly higher than the measurements which will affect the development of the size distribution. Further, it seems that the differences between the measurements and the model simulation increase with time. Furthermore, the differences are largest at in between the two modes of the size distribution. That is not discussed at all. I would assume that this is caused by an uncertainty in the model. Is the coagulation really correctly calculated? Can't you play around a little with the model and check if the differences get larger/smaller than the nucleation rate is lower/higher (assuming according lower/higher concentrations of dimethylamine and sulfuric acid)?

Even larger are the difference between the aerosol volumes, but from the discussion it sounds as that the agreement is perfect which is obviously not true.

P12, l440: Same here as for the Almeida et al. (2013) study, add some more information how Jen et al. (2016) derived their nucleation rates. Was it a similar experiment as the one you performed. If no, what has been done differently etc.

P12-13, l439-478: This text part definitely belong to the method section than to the result section.

P13, l481: The abbreviation DMA has not been introduced yet. I would suggest to keep writing dimethylamine throughout the manuscript. Otherwise the text is quite difficult to read with all the abbreviations that are already used.

P15, l529 and l530: Please give the abundances of $H_2SO_4$ and dimethylamine as concentrations and mixing ratios.

P15, l550: I would suggest to write: "This study confirms the results derived in previous studies."

P15, l553, l575, l576: please give the according concentrations and mixing ratios.

P15, l605: It should read "numerical model".

P16, l608-609: units for M and W missing or are these dimensionless?

P30, Figure 3 caption: add DMA in brackets after dimethylamine. Give the abundances for $H_2SO_4$ and dimethylamine in both concentration and mixing ratio.

---

## Referee Comment (RC2) · Anonymous Referee #1 · 13 Oct 2017

This paper presents revised calculations of nucleation rate of CLOUD7 ternary nucleation of sulfuric acid-dimethyl amine-water (278 K, 38% RH, sulfuric acid concentration between 1e6 and 3e7 cm-3 and dimethylamine mixing ratio of ∼40 pptv; shown in Almeida et al., Nature 2013), and concludes that under this base-dominant and low temperature conditions, DMA-THN takes place in kinetic regime, that is, collision-limited coagulation of clusters (without nucleation barrier; and no effects of evaporation rates for H2SO4-DMA). The conditions with high concentrations of DMA and low temperatures seem to be plausible for barrier-less nucleation, although it is still difficult

to conclude this with limited knowledge of thermodynamics of nucleation (e.g., BHN, THN or IIN). This recalculation is useful to the community. I suggest to tone down other conclusions and remove the simulation of atmospheric NPF with low amines (Section 3.6), as described below.

The authors conclude that the CLOUD7 results are consistent with Jen et al. ACP 2016 flow tube THN experiments; the latter was undertaken at a high temperature and acidic conditions (6e9 cm-3, tens of pptv of amines, and near 300K). If they both take place via the same collision-limited coagulation processes, this is most likely because of very different reasons. For CLOUD7, this is due to low temperature and high DMA. And for Jen et al., this is due to exceedingly high sulfuric acid (so that nucleation rates are sensitively dependent on base concentrations). To show they are consistent, the best way is to use the current nucleation algorithm to re-calculate nucleation rates using the experimental data from Jen et al. If even 0,1 pptv DMA makes nucleation kinetic (at both acidic and basic conditions and both low and high temperatures), then di-amines (Jen et al., GRL 2016) should not further enhance nucleation rates, which is not the case. Also, ammonia and amines also should not enhance nucleation (Yu et al., GRL 2012; Glasoe et al., JGR 2015). So, this is an overstatement: "using this model, the findings from the present study and the flow tube experiment can be brought into good agreement."

The authors also conclude that in the boundary layer (temperature > 245K), even with 0.1 ppt level of dimethylamine, nucleation would proceed with the collision limited process. The section 3.6 is too speculative and should be removed – see below minor comments in detail, to improve the paper quality.

Note, DMA is the only amine that so far CLOUD used and published, but there are other amines that can be as effective as DMA as ternary species, such as trimethylamine (Yu et al., GRL 2012; Glasoe JGR 2015; Jen GRL 2016; Hanson et al., JPC 2017), diamines (Jen GRL 2016) and even methylamine (Chen et al., EST 2017; Chen et al., JPC 2016). And these amines are present in almost everywhere in our environment at

[Figure]

anytime, especially within the boundary layer.

Also, some assumptions used in this study have apparent limitations (in addition to evaporation rates at 247K). For example. RH has significant effects on both nucleation and growth rates of sub-3 nm particles, as shown by flow tube experiments, even within a wide range of temperatures covering both CLOUD7 and Jen et al. conditions (Yu et al., 2014). Yu et all also showed that growth rates are not constant within the sub-3 nm particle size.

Please provide detailed tables of variables/values/sources used in models in supporting material, including evaporation rates for clusters.

Please remove redundant sentences.

Line 66: At the surface level, in fact sulfuric acid can be as high the conditions of CLOUD7 (very frequently), and amines/ammonia are abundant (see above). Rather, the problem is high temperatures and high surface area. The question is under this conditions, the very low 0.1 ppt of DMA can make nucleation proceeding kinetically, without any other species?

Line 82: please cite You and Lee, EC, 2012; You et al., ACP 2014.

Line 88: clarify that Kirkby et al. Nature 2016 conclusion is based on CLOUD chamber studies, and this yet needs to be verified by atmospheric measurements, in pristine forests during the night, for example.

Line 137: indicate the detection limit and time resolution of the IC method used to detect amines.

Line 148: "time-rate-of-change"?

Line 264: si,j?

Lines 320-327: what is the exact sulfuric acid background level (without OH)? Is it dependent on SO2 or temperature? Why do you have to discount that sulfuric acid?

[Figure]

How did you know that is not "real" sulfuric acid?

Line 474: ke,A1b1= 0.1, 1, 10(?) s-1?

Line 481: 40 or 20 ppt? (earlier it was mentioned as 20 ppt).

Line 497: 1 ppt DMA is still larger than 5e6 cm-3 sulfuric acid, so this is a base dominant environment. So, this is again quite different from the Jen-ACP-2016 condition.

Line 507: Why would you assume that Hytiala has low DMA around 0.1 ppt, because CI-Api-TOF did not measured DMA? Remove.

Line 517: CS 2e-3 s-1 is very clean, compared to most of boundary layer conditions.

Line 527-535: why assume DMA is anti-correlated with OH (due to OH oxidation)? In fact, atmospheric measurements, even by the authors (Kurten et al., ACP 2016; Jen et al., GRL 2017) and others (You et al., ACP 2014; Yao et al., ACP 2016), consistently showed that amines have the same diurnal cycles as ambient temperatures, higher concentrations during the day than at night. This indicates that the main sink of amines in the atmosphere is condensation to aerosols, and not the oxidation by OH or photolysis (You et al., ACP 2014).

Lines 535-541: remove.

Lines 599-560: reword this conclusion here and at other places.

Line 560 and on: Please see Yu et al., JGR 2017 on RH effects on J and GR for sub-3 nm particles. Please cite this paper.

Line 564 and on: Please see Hanson et al., JPC 2017. Evaporation rates are highly dependent on thermodynamics data. Cite this.

Line 641: Zhao et al., 2014 – if I recall correctly, this cited study intentionally included excessively high sulfuric acid in the inlet of CIMS to see SA-DMA clusters, rather than directly measure the "existing" SA-DMA clusters from ambient air. (This is very similar

to Jen et al. flow tube environment, where acid exceeds base. Interesting instrumentation mechanics, if compare cluster-CIMS vs. CI-TOF ?)

Chen, H. and B.J. Finlayson-Pitts, New Particle Formation from Methanesulfonic Acid and Amines/Ammonia as a Function of Temperature. Environmental Science & Technology, 2017. 51: p. 243-252.

Chen, H., M.E. Varner, R.B. Gerber, and B.J. Finlayson-Pitts, Reactions of Methanesulfonic Acid with Amines and Ammonia as a Source of New Particles in Air. The Journal of Physical Chemistry B, 2016. 120: p. 1526-1536.

Hanson, D.R., I. Bier, B. Panta, C.N. Jen, and P.H. McMurry, Computational Fluid Dynamics Studies of a Flow Reactor: Free Energies of Clusters of Sulfuric Acid with NH3 or Dimethyl Amine. The Journal of Physical Chemistry A, 2017. 121: p. 3976-3990.

Kürten, A., A. Bergen, M. Heinritzi, M. Leiminger, V. Lorenz, F. Piel, M. Simon, R. Sitals, A.C. Wagner, and J. Curtius, Observation of new particle formation and measurement of sulfuric acid, ammonia, amines and highly oxidized organic molecules at a rural site in central Germany. Atmos. Chem. Phys., 2016. 16: p. 12793-12813.

Yu, H., L. Dai, Y. Zhao, V.P. Kanawade, S.N. Tripathi, X. Ge, M. Chen, and S.-H. Lee, Laboratory observations of temperature and humidity dependencies of nucleation and growth rates of sub-3 nm particles. Journal of Geophysical Research: Atmospheres, 2017. 122: p. 1919-1929.

---

## Referee Comment (RC3) · Anonymous Referee #3 · 6 Nov 2017

In this study, formation rates published by Almeida et al. (2013) for ternary sulfuric acid (SA) nucleation with dimethylamine (DMA) in the CLOUD chamber are re-analyzed with a method that takes into account self-coagulation. The authors argue that particle formation rates at 1.7 nm are more than a factor of 10 higher than those reported by Almeida et al. (2013), which would imply that SA-DMA new particle formation is significant at lower DMA gas-phase concentrations than previously thought. The revised formation rates agree well with rates calculated by a kinetic aerosol model at different particle diameters. Therefore, the authors conclude that nucleation for the conditions

studied here proceeds at rates that are collision-controlled.

General comments:

I think this manuscript is well written and contains some interesting results and conclusions that makes it suitable for publication in ACP. However, since the manuscript focuses mainly on a re-evaluation of particle formation rates from the paper by Almeida et al. (2013), I think more information needs to be given on the approach used by Almeida et al. for extrapolating their formation rates. I suggest that the authors add a schematic diagram or a table illustrating how 1) Almeida et al. have calculated their formation rates and 2) how the authors of the present study have calculated their formation rates. Such a diagram should also include information on what instruments were used when deriving the particle formation rates, and the necessary corrections. For instance, the authors state on lines 335-338 that Almeida et al. (2013) made an extrapolation from 3 to 1.7 nm when deriving their formation rates at 1.7 nm. How was this extrapolation done? Furthermore, the authors of the present study use data from the smallest SMPS size channel to calculate the formation rate. As the authors admit on lines 344-345, "the smallest SMPS size channels need to be corrected by large factors to account for losses and charging probability, which introduces uncertainty". How were these corrections made, and how large were the corrections relative to the actual measured number concentrations? In addition, the authors assume on line 366 that the growth rate is independent of size which adds more uncertainty. How large are these uncertainties compared to the "error" resulting from the extrapolation method used by Almeida et al. (2013)?

Another general comment I have is related to the fact that there is another recent study focusing on nanoparticle growth for the SA-DMA system in the CLOUD chamber by Ahlm et al. (2016), where most authors of this manuscript were co-authors. In that study, model simulations and measurements with three different instruments indicated an increasing particle-phase DMA/SA molar ratio with increasing particle size due to a decreasing Kelvin-effect of DMA with increasing size, from ~1.5 to 20 nm. The results

of that study appear, at least to this reviewer, to be inconsistent with the view provided in this manuscript that nucleation and growth up to ∼80 nm are completely collision-controlled. I think there needs to be some explanation, or at least, discussion of this issue.

Specific comments:

1. In the Almeida et al. (2013) paper, the ACDC model reproduced ternary SA-ammonia formation rates almost perfectly, but somewhat overpredicted ternary SA-DMA formation rates compared to observations in the CLOUD chamber. I think it could be worth mentioning that the conclusion within this manuscript, that ternary SA-DMA formation rates in the CLOUD chamber were underestimated by Almeida et al., brings the formation rates much closer to predictions by the ACDC model.

2. Sect. 2.1: Please describe the SMPS measurements including corrections.

3. Line 35: The word "advanced" is not very useful for the reader. It is better to try to explain as clearly as possible the difference between the approach used here and the method used by Almeida et al.

4. Line 40: "modeled and measured size distributions show good agreement". I think it should be mentioned that this was for one nucleation event that you studied in detail, unless you have analyzed other events as well.

5. Lines 137-138: To what extent was dimethylamine oxidized by OH within the chamber during these events? Were any oxidation products detected and may these have contributed to new particle formation?

6. Line 321: How high is "relatively high", and how do the authors know there was no sulfuric acid in the chamber? Do the authors think this is a general problem with using a CIMS for measuring sulfuric acid?

References

[Figure]

Ahlm et al. (2016). Aerosol Sci. Technol., 50, 1017-1032, 2016. Almeida et al. (2013). Nature, 502, 359-365.

---

## Author Comment (AC1) · 24 Nov 2017

We thank the referee for the constructive comments, which are added in full below (in black font). Our replies are given in blue font directly after the comments; text that has been added to the manuscript is shown in red font.

**Anonymous Referee #1**

This paper presents revised calculations of nucleation rate of CLOUD7 ternary nucleation of sulfuric acid-dimethyl amine-water (278 K, 38% RH, sulfuric acid concentration between 1e6 and 3e7 cm$^{-3}$ and dimethylamine mixing ratio of ~40 pptv; shown in Almeida et al., Nature 2013), and concludes that under this base-dominant and low temperature conditions, DMA-THN takes place in kinetic regime, that is, collision-limited coagulation of clusters (without nucleation barrier; and no effects of evaporation rates for H$_2$SO$_4$-DMA). The conditions with high concentrations of DMA and low temperatures seem to be plausible for barrier-less nucleation, although it is still difficult to conclude this with limited knowledge of thermodynamics of nucleation (e.g., BHN, THN or IIN). This recalculation is useful to the community. I suggest to tone down other conclusions and remove the simulation of atmospheric NPF with low amines (Section 3.6), as described below.

1) The authors conclude that the CLOUD7 results are consistent with Jen et al. ACP 2016 flow tube THN experiments; the latter was undertaken at a high temperature and acidic conditions (6e9 cm$^{-3}$, tens of pptv of amines, and near 300 K). If they both take place via the same collision-limited coagulation processes, this is most likely because of very different reasons. For CLOUD7, this is due to low temperature and high DMA. And for Jen et al., this is due to exceedingly high sulfuric acid (so that nucleation rates are sensitively dependent on base concentrations). To show they are consistent, the best way is to use the current nucleation algorithm to re-calculate nucleation rates using the experimental data from Jen et al. If even 0.1 pptv DMA makes nucleation kinetic (at both acidic and basic conditions and both low and high temperatures), then di-amines (Jen et al., GRL 2016) should not further enhance nucleation rates, which is not the case. Also, ammonia and amines also should not enhance nucleation (Yu et al., GRL 2012; Glasoe et al., JGR 2015). So, this is an overstatement: "using this model, the findings from the present study and the flow tube experiment can be brought into good agreement."

First of all, we would like to clarify some of the statements made in the comment:

We do not claim that nucleation is collision-controlled for all conditions of the Jen et al. studies. The reviewer is correct that in some cases (especially at low base to acid ratios), diamines yield even higher formation rates, compared with the amines. This observation alone indicates that sulfuric acid-dimethylamine new particle formation is not entirely collision-controlled for all conditions.

The CLOUD data and the model inter-comparison show, however, that nucleation can proceed at rates that are compatible with collision-controlled nucleation. This is due to the fact that the dimethylamine mixing ratio is ~100 times higher (40 pptv, i.e. $1\times10^9$ cm$^{-3}$) compared with the highest sulfuric acid concentration (~$1\times10^7$ cm$^{-3}$) in CLOUD. Under these conditions, the modeled cluster concentrations are essentially insensitive to the use of non-zero evaporation rates as long as these are as small as reported by Jen et al. (2016a). This is explained in section 2.5 of the manuscript, where the evaporation rates are listed.

However, when using a low DMA mixing ratio (0.1 pptv), the modeled new particle formation rates (including the evaporation rates from Jen et al., 2016a) are significantly lower than for collision-controlled nucleation (by about a factor of ~100, see Fig. 3, lower panel).

It is true however, that we have not shown yet that our model can replicate all of the flow tube results by Jen et al. (2014, 2016a, 2016b). Still, qualitatively the studies agree very well. This can, e.g., be seen from the experiments by Jen et al. (2016b) for amines and diamines. At high base to acid ratio particle formation reaches a plateau value that is similar for all the different bases. This shows that eventually, the new particle formation rates are indistinguishable from collision-controlled nucleation. Only at low base to acid ratio ($< ~0.5$) particle formation decreases with lower base concentrations. For these conditions, the diamines studied by Jen et al. (2016b) can actually lead to even more efficient NPF compared with DMA. However, this can probably be explained by even lower evaporation rates for some of the clusters that can still evaporate at slow rates in the sulfuric acid-DMA system ($k_{e,A1B1} = 0.1$ s$^{-1}$, $k_{e,A3B1} = 1$ s$^{-1}$, $k_{e,A3B2} = 1$ s$^{-1}$, $k_{e,A4B1} = \infty$ s$^{-1}$, see section 2.5). The same can be true for the synergistic effects (interaction between amines and NH$_3$) reported by Glasoe et al. (2015): additional stabilization of some clusters can occur that are still not entirely stable for pure sulfuric acid-DMA nucleation at low base to acid ratio.

These points are now explained in more detail in section 3.5:

"The conditions are only such that, due to the high DMA mixing ratio, most of the clusters (including the monomer) probably contain as many DMA molecules as sulfuric acid molecules; this results in very stable cluster configurations (Ortega et al., 2012). When DMA mixing ratios are low, most sulfuric acid clusters contain, however, only a small number of DMA molecules. As these clusters can evaporate more rapidly, the overall formation rate is slowed down (Ortega et al., 2012; Hanson et al., 2017). For low base to acid ratios, it can therefore matter whether a cluster is stabilized by a dimethylamine, a diamine (Jen et al., 2016) or by both an amine and an ammonia molecule (Glasoe et al., 2015). This can explain the more efficient NPF due to diamines or the synergistic effects involving amines and ammonia at low base to acid ratios. At high base to acid ratios, the differences in the effective evaporation rates become small (Jen et al., 2016b)."

With our model we have not attempted to recalculate all of the Jen et al. (2014, 2016a, 2016b), Glasoe et al. (2015) and Hanson et al. (2017) results as this would be beyond the scope of our manuscript. Rather than this, a comparison is now performed with a formula presented by Hanson et al. (2017) that summarizes their results on sulfuric acid-DMA nucleation from the flow tube studies. This formula, i.e.,

$$J_{1.4nm} = exp\left(-129 + \frac{16200\ K}{T}\right) \cdot \left(\frac{N_1}{cm^{-3}}\right)^3 \cdot \left(\frac{DMA}{cm^{-3}}\right)^{1.5}$$

is provided in the revised manuscript (new equation (10) in section 3.3) and a comparison between its values and the results from the present study is shown in Fig. 1 and Fig. 3, lower panel.

In addition to the changes mentioned above, to address the reviewers concern, we have attempted to highlight that the good agreement between our measurements, the model results and the flow tube study is so far only found for the conditions of high base to acid ratios (and DMA). These changes are mentioned in the following.

> We have changed the statement in the abstract

"Using this model, the findings from the present study and the flow tube experiment can be brought into good agreement."

to

"Using this model, the findings from the present study and the flow tube experiment can be brought into good agreement for the high base to acid ratios (~100) relevant for this study."

> Section 4:

"Even when evaporation rates for the less stable clusters are introduced in the model (Jen et al., 2016a) the resulting particle formation rates are effectively indistinguishable from the kinetic model results for CLOUD7 conditions"

Changed to:

"Even when evaporation rates for the less stable clusters are introduced in the model (Jen et al., 2016a) the resulting particle formation rates are effectively indistinguishable from the kinetic model results for CLOUD7 conditions (i.e., at the high dimethylamine to acid ratio of ~100)."

> Section 5:

"This indicates that the data from the flow tube study by Jen et al. (2016a) and from CLOUD (Kürten et al., 2014) are consistent."

Changed to:

"This indicates that the data for sulfuric acid-dimethylamine from the flow tube study by Jen et al. (2016a) and from CLOUD (Kürten et al., 2014) are consistent for the high base to acid ratio relevant for this study (dimethylamine to sulfuric acid monomer ratio of ~100)."

2) The authors also conclude that in the boundary layer (temperature > 245K), even with 0.1 pptv level of dimethylamine, nucleation would proceed with the collision limited process. The section 3.6 is too speculative and should be removed – see below minor comments in detail, to improve the paper quality.

After including the calculated formation rates from a recently published study by Hanson et al. (2017) in Fig. 1 and Fig. 3, lower panel, the following became evident:

The Hanson et al. (2017) equation (now also included in the manuscript, see equation (10) in section 3.3) is predicting lower NPF rates for the small DMA mixing ratios compared with our

model. This indicates that there exists some uncertainty for the low DMA mixing ratios regarding nucleation as no experiments have been made at mixing ratios below 1 pptv.

Therefore, we agree with the referee and have removed Fig. 4 together with the corresponding discussion (section 3.6) from the manuscript.

3) Note, DMA is the only amine that so far CLOUD used and published, but there are other amines that can be as effective as DMA as ternary species, such as trimethylamine (Yu et al., GRL 2012; Glasoe JGR 2015; Jen GRL 2016; Hanson et al., JPC 2017), diamines (Jen GRL 2016) and even methylamine (Chen et al., EST 2017; Chen et al., JPC 2016). And these amines are present in almost everywhere in our environment at anytime, especially within the boundary layer.

We agree with this comment. From what we found from the literature, DMA and TMA behave very similar in terms of nucleation and the tested diamines (ethylene diamine, tetramethylethylene diamine and butanediamine/putrescine) seem to be at least as efficient (Jen et al., 2016b). These substances have been measured at mixing ratios above several pptv and therefore it is a very important question to what extent they are responsible for new particle formation in the atmosphere. We hope that our manuscript can stimulate further research in this direction.

4) Also, some assumptions used in this study have apparent limitations (in addition to evaporation rates at 278 K). For example, RH has significant effects on both nucleation and growth rates of sub-3 nm particles, as shown by flow tube experiments, even within a wide range of temperatures covering both CLOUD7 and Jen et al. conditions (Yu et al., 2017). Yu et al. also showed that growth rates are not constant within the sub-3 nm particle size.

It is true that RH can have a significant effect on new particle formation rates (e.g. Duplissy et al., 2016, etc.). However, the mentioned study by Yu et al. (2017) reported results for the binary system of sulfuric acid and water; base molecules were only present at contaminant level ($NH_3$ < 23 pptv, methylamine < 1.5 pptv and dimethylamine < 0.52 pptv). The influence of RH on ternary nucleation (involving sulfuric acid, water and $NH_3$ or amines) is far less studied. However, a recent study based on quantum chemical calculations indicates that RH has only a very small effect on new particle formation rates (only a factor of less than 1.5 over the range of 0 to 100% RH) for dimethylamine (Olenius et al., 2017).

Regarding the growth rates, the study by Yu et al. (2017) showed that the particle growth rate does not change significantly over the range from ~1.7 to 2.2 nm (Fig. 1 in Yu et al., 2017). For larger particles, no data were shown in their publication. However, the study by Kürten et al. (2015a) investigated the size dependency of the growth rates for collision-controlled nucleation; no significant size dependency was found within the size range for 1.7 to 3.2 nm.

The growth rate does however change with temperature and relative humidity. This can have several reasons:

– For very low temperatures (or very stable clusters), nucleation will approach the collision-controlled situation. Under such conditions, a significant contribution to growth from clusters is expected (Lehtipalo et al., 2016).

- The base contaminant can increase with higher RH as the contaminants can originate from the water supply or because of wall effects where water displaces base molecules from the chamber or flow reactor walls (e.g. Vaitinen et al., 2014).

- Additional water molecules lead to faster particle growth at higher RH because the water is brought in with the condensing sulfuric acid as sulfuric acid includes more water ligands at increasing RH (Hanson and Eisele, 2000).

While all the factors can contribute to accelerated growth at varying conditions, they indicate nothing about the size-dependency of the growth rate. As stated earlier, the growth rate size-dependency seems to be relatively weak for collision-controlled nucleation. As the data from the present study are consistent with collision-controlled nucleation and new particle formation for the sulfuric acid-dimethylamine-system has been reported to be almost insensitive to RH (Olenius et al., 2017), only brief information about RH effects has been added (to section 4):

"Water could play a role at higher relative humidities, although quantum chemical calculations suggest that it plays only a minor role in NPF for the system of sulfuric acid and dimethylamine (Olenius et al., 2017); this contrasts the sulfuric acid-water system (see e.g. Zollner et al. 2012; Duplissy et al., 2016; Yu et al., 2017)."

Possible effects of water leading to a shift in the particle size distribution are mentioned also in section 4.

5) Please provide detailed tables of variables/values/sources used in models in supporting material, including evaporation rates for clusters.

In the light of this comment, the equations from section 2.5 were moved to an appendix (Appendix A). Furthermore, a new table (Table 1) was added to the manuscript; this table indicates the evaporation rates and for what model calculations they were included.

6) Please remove redundant sentences.

It is not clear to which sentences this comment is referring to. However, in the context of other comments some statements were removed or rewritten. We hope that this adequately addresses the reviewers request.

7) Line 66: At the surface level, in fact sulfuric acid can be as high as the conditions of CLOUD7 (very frequently), and amines/ammonia are abundant (see above). Rather, the problem is high temperatures and high surface area. The question is under these conditions, the very low 0.1 pptv of DMA can make nucleation proceeding kinetically, without any other species?

As stated above (reply to comment 2), we do not claim that new particle formation is kinetic at DMA = 0.1 pptv. In addition, the evaporation rates used in the present study were derived for temperatures at ~300 K (Jen et al., 2016a), therefore, they should well represent the conditions for relatively warm conditions (see also discussion in section 4 of the manuscript).

Regarding the condensation sink, the reviewer is correct. The conditions for the simulations shown in Figure 4 (removed, see comment 2)) are rather clean (condensation sink of $2\times10^{-3}$ s$^{-}$

[1]). However, the measured condensation sink for the boreal forest in Hyytiälä/Finland are close to this value (see also reply to comment 19)). For a higher condensation sink, the expected new particle formation rates would be reduced and it is possible that this can explain the absence of nucleation even when amine mixing ratios are relatively high.

However, rather than depleting the growing clusters, the condensation sink can also have the effect of depleting the amines. Kürten et al. (2016b) have observed that the amine mixing ratios can be reduced by up to a factor 5 during new particle formation events compared to days when no nucleation is observed. As amines are not produced in the gas phase (unlike sulfuric acid), their clustering with sulfuric acid monomers and small sulfuric acid clusters/particles very likely can lead to significant reduction in the amine mixing ratios. This would indicate that new particle formation involving amines in the atmosphere could be self-limiting, i.e., after an initial burst of particles, new particle formation could be slowed down soon after when amine mixing ratios decrease. This effect could most strongly be caused by the newly formed clusters and particles that can significantly contribute to the condensation sink. However, the CS is most often determined from size-distribution measurements starting above ~3 nm and therefore does not include the newly formed clusters and smallest particles.

Since the section showing the atmospheric simulations has been removed (see comment 2 above) a short summary of this effect is added to the conclusion section (section 5):

"High time resolution (several minutes or better) for the amine measurements during nucleation events is also important. This can show, whether amines can be significantly depleted during NPF. As amines are not produced in the gas phase (unlike sulfuric acid), their clustering with sulfuric acid monomers and small sulfuric acid clusters/particles very likely can lead to a significant reduction in the amine mixing ratios (Kürten et al., 2016b). This would indicate that new particle formation involving amines in the atmosphere could be self-limiting, i.e., after an initial burst of particles, new particle formation could be slowed down soon after when amine mixing ratios decrease."

8) Line 82: please cite Yu and Lee, EC, 2012; You et al., ACP 2014.

Done.

9) Line 88: clarify that Kirkby et al. Nature 2016 conclusion is based on CLOUD chamber studies, and this yet needs to be verified by atmospheric measurements, in pristine forests during the night, for example.

The sentence was modified as follows to clarify that the Kirkby et al. (2016) study is based on chamber experiments:

"These highly-oxygenated molecules have been found to nucleate efficiently in a chamber study even without the involvement of sulfuric acid, especially when ions take part in the nucleation process (Kirkby et al., 2016)."

10) Line 137: indicate the detection limit and time resolution of the IC method used to detect amines.

The sentence was changed to include the requested information:

"The mixing ratio of dimethylamine was determined by ion chromatography with a detection limit of 0.2 to 1 pptv at a time resolution between 70 and 210 minutes (Praplan et al., 2012; Simon et al., 2016)."

11) Line 148: "time-rate-of-change"?

The expression "time-rate-of-change" was replaced by "time derivative".

12) Line 264: $s_{i,j}$?

The factor $s_{i,j}$ is 0.5 when $i = j$ and 1 otherwise. It is explained at the end of section 2.2.

13) Lines 320–327: what is the exact sulfuric acid background level (without OH)? Is it dependent on $SO_2$ or temperature? Why do you have to discount that sulfuric acid?

For the chemical system relevant for the present study ($SO_2$, $O_3$, $H_2O$ and DMA without the presence of UV light) we have no evidence for significant dark production of sulfuric acid. Therefore, we consider any measured $H_2SO_4$ at zero UV as instrumental background. This follows also from a direct comparison between the independently calibrated nitrate CIMS (Kürten et al., 2011; Kürten et al., 2012) and nitrate CI-APi-TOF (Kürten et al., 2014). When UV light produces significant $H_2SO_4$ both instruments agree quite well, whereas at zero UV the CIMS showed significantly higher [$H_2SO_4$] compared to the CI-APi-TOF during the DMA experiments. For this reason, it is justified to subtract the CIMS background from the concentrations measured during periods with activated UV light.

The sentence in the last paragraph of section 3.1 was changed to indicate that the CIMS background was an instrumental artifact:

"However, taking into account a subtraction of this instrumental background (reaching sometimes values above $1 \times 10^6$ cm$^{-3}$) leads to a shallower slope for $J_{1.7nm}$ vs. sulfuric acid and brings the corrected CIMS values in a good agreement with the sulfuric acid measured by the CI-APi-TOF."

14) How did you know that is not "real" sulfuric acid?

See reply to previous comment.

15) Line 474: $k_{e,A1B1}$ = 0.1, 1, 10(?) s$^{-1}$?

We thank the reviewer a lot for realizing this mistake. The sentence should read:

"The evaporation rates considered are $k_{e,A1B1}$ = 0.1 s$^{-1}$, $k_{e,A3B1}$ = 1 s$^{-1}$ and $k_{e,A3B2}$ = 1 s$^{-1}$ (Jen et al., 2016a)."

16) Line 481: 40 or 20 pptv? (earlier it was mentioned as 20 pptv).

Earlier it was mentioned that DMA was always present at 20 pptv or higher. 40 pptv are an average mixing ratio.

17) Line 497: 1 pptv DMA is still larger than 5e6 cm$^{-3}$ sulfuric acid, so this is a base dominant environment. So, this is again quite different from the Jen-ACP-2016 condition.

It is true that 1 pptv (= ca. $2.5\times10^7$ cm$^{-3}$) of DMA is higher than $5\times10^6$ cm$^{-3}$ of sulfuric acid; therefore, the reviewer is correct that this condition can still be considered base-dominated.

The discussion about Fig. 3, lower panel, includes now the data from Hanson et al. (2017). Their equation was included to the manuscript (new equation (10)) and corresponding data were added to Fig. 1 and Fig. 3 (lower panel). In addition, the statements about the agreement between the flow tube and the CLOUD studies were revised.

18) Line 507: Why would you assume that Hyytiälä has low DMA around 0.1 pptv, because CI-APi-TOF did not measured DMA? Remove.

Sipilä et al. (2015) detected no DMA above the detection limit (0.12 pptv) of their instrument in Hyytiälä. Therefore, a mixing ratio of ~0.1 pptv can be regarded as an upper limit for this site.

A very recent study (Hemmilä et al., 2017) reported new amine measurements from Hyytiälä/Finland. While DMA was below the detection limit of the instrument (ca. 0.2 pptv), on some days up to ~3 pptv were measured in the gas phase. For trimethylamine, a monthly average of 0.1 to 0.2 pptv was reported. In the particle phase, the monthly averages ranged from around 0.5 to 4 pptv. These numbers can be taken as evidence that the mixing ratios for DMA and TMA are non-zero in Hyytiälä/Finland – at least on some days – and that their contribution to new particle formation should be considered. An earlier study from Mäkela et al. (2001) found an enrichment of DMA in particles during nucleation events.

However, as mentioned before in response to comment (2) section 3.6 was removed.

19) Line 517: CS = $2\times10^{-3}$ s$^{-1}$ is very clean, compared to most of boundary layer conditions.

It is true that this condensation sink is rather low but it is representative of the environment for which this model study was performed. Data shown by Dada et al. (2017) indicate a condensation sink which is on average ~$2\times10^{-3}$ s$^{-1}$ during new particle formation event days in Hyytiälä/Finland. As section 3.6 was removed, this is, however, not further discussed in the manuscript.

20) Line 527–535: why assume DMA is anti-correlated with OH (due to OH oxidation)? In fact, atmospheric measurements, even by the authors (Kürten et al., ACP 2016; Jen et al., GRL 2017) and others (You et al., ACP 2014; Yao et al., ACP 2016), consistently showed that amines have the same diurnal cycles as ambient temperatures, higher concentrations during the day

than at night. This indicates that the main sink of amines in the atmosphere is condensation to aerosols, and not the oxidation by OH or photolysis (You et al., ACP 2014).

In line 527 we suggest that DMA can be depleted by the newly formed particles. OH oxidation would be another possibility (line 525/526). The observed diurnal cycle of amines (higher during the day) can have several reasons, e.g., stronger emissions due to elevated temperature, or some repartioning of condensed amines from the aerosol to the gas phase. However, this would be the case for the unperturbed atmosphere (without nucleation). If new particles are formed (containing sulfuric acid), these should act as an additional sink for the amines, which could bind to the growing acidic particles. The loss rate of DMA molecules on a sulfuric acid dimer alone is ca. $1 \times 10^{-4}$ s$^{-1}$ (product between the collision rate, $10^{-9}$ cm$^3$ s$^{-1}$, and a sulfuric acid dimer concentration of $1 \times 10^5$ cm$^{-3}$, see Kürten et al., 2016b). Considering the total loss rate of DMA on nucleating clusters, would correspondingly increase the condensation sink for DMA significantly. Therefore, new particle formation should lead to some depletion of amines, if their mixing ratio does not strongly exceed the sulfuric acid concentration.

In addition, the mentioned publications (You et al., 2014; Kürten et al., 2016b; Yao et al., 2016) showed no clear correlation between temperature (or a clear daily pattern for most of the amines). In fact, the Yao et al. (2016) study showed a maximum for the C2-amines in the morning, which would actually be consistent with the consumption of amines by new particle formation. The other studies (You et al., 2014; Kürten et al., 2016b) showed no significant variation of any of the amines, except for the C4 and C6 amines, which peaked during mid-day. Since these data, however, showed averages over many days including days with and without nucleation it is hard to draw any solid conclusions.

21) Lines 535-541: remove.

The whole section 3.6 was removed (see comment 2 above); therefore, this comment is obsolete.

22) Lines 559–560: reword this conclusion here and at other places.

The whole sentence was deleted.

23) Line 560 and on: Please see Yu et al., JGR 2017 on RH effects on J and GR for sub-3 nm particles. Please cite this paper.

As mentioned before (reply to comment 4) we do not think that RH has a very strong effect on the new particle formation and growth rates for the conditions of the present study (sulfuric acid-dimethylamine system).

However, we have changed the paragraph in section 4 as follows:

"Water could play a role at higher relative humidities, although quantum chemical calculations suggest that it plays only a minor role in NPF for the system of sulfuric acid and dimethylamine (Olenius et al., 2017); this contrasts the sulfuric acid-water system (see e.g. Zollner et al. 2012; Duplissy et al., 2016; Yu et al., 2017). In addition, it is not exactly known how temperature influences the cluster evaporation rates (Hanson et al., 2017)."

24) Line 564 and on: Please see Hanson et al., JPC 2017. Evaporation rates are highly dependent on thermodynamics data. Cite this.

Done (see reply to comment 23).

25) Line 641: Zhao et al., 2014 – if I recall correctly, this cited study intentionally included excessively high sulfuric acid in the inlet of CIMS to see SA-DMA clusters, rather than directly measure the "existing" SA-DMA clusters from ambient air. (This is very similar to Jen et al. flow tube environment, where acid exceeds base. Interesting instrumentation mechanics, if compare cluster-CIMS vs. CI-TOF?)

It is true that some measurements in the Zhao et al. (2011) study were made when $H_2SO_4$ was added to the cluster-CIMS inlet. However, measurements were also made without the addition of $H_2SO_4$. The observed signals during these measurements were still consistent with the presence of neutral sulfuric acid amine cluster.

**References**

Almeida, J., et al.: Molecular understanding of sulphuric acid-amine particle nucleation in the atmosphere, *Nature*, 502, 359–363, doi: 10.1038/nature12663, 2013.

Chen, H., and Finlayson-Pitts, B. J.: New Particle Formation from Methanesulfonic Acid and Amines/Ammonia as a Function of Temperature, *Env. Science Technol.*, 51, 243–252, 2017.

Chen, H., et al.: Reactions of Methanesulfonic Acid with Amines and Ammonia as a Source of New Particles in Air, *J. Phys. Chem. B*, 120, 1526–1536, 2016.

Dada, L., et al.: Long-term analysis of clear-sky new particle formation events and nonevents in Hyytiälä, *Atmos. Chem. Phys.*, 17, 6227–6241, doi: 10.5194/acp-17-6227-2017, 2017.

Duplissy, J., et al.: Effect of ions on sulfuric acid-water binary particle formation II: Experimental data and comparison with QC-normalized classical nucleation theory, *J. Geophys. Res.-Atmos.*, 121, 1752–1775, doi: 10.1002/2015JD023539, 2016.

Glasoe, W. A., et al.: Sulfuric acid nucleation: An experimental study of the effect of seven bases, *J. Geophys. Res.-Atmos.*, 120, 1933–1950, doi: 10.1002/2014JD022730, 2015.

Hanson, D. R., and Eisele, F.: Diffusion of $H_2SO_4$ in humidified nitrogen: Hydrated $H_2SO_4$, *J. Phys. Chem. A*, 104, 1715–1719, doi: 10.1021/jp993622j, 2000.

Hanson, D. R., et al.: Computational Fluid Dynamics Studies of a Flow Reactor: Free Energies of Clusters of Sulfuric Acid with $NH_3$ or Dimethyl Amine, *J. Phys. Chem. A*, 121, 3976–3990, doi: 10.1021/acs.jpca.7b00252, 2017.

Hemmilä, M., et al.: Amines in Boreal Forest Air at SMEAR II Station in Finland, *Atmos. Chem. Phys. Discuss.*, doi: 10.5194/acp-2017-958, in review, 2017.

Jen, C. N., et al.: Chemical ionization of clusters formed from sulfuric acid and dimethylamine or diamines, *Atmos. Chem. Phys.*, 16, 12513–12529, doi: 10.5194/acp-16-12513-2016, 2016a.

Jen, C. N., et al.: Diamine-sulfuric acid reactions are a potent source of new particle formation, *Geophys. Res. Lett.*, 43, 867–873, doi: 10.1002/2015GL066958, 2016b.

Kirkby, J., et al.: Ion-induced nucleation of pure biogenic particles, *Nature*, 533, 521–526, doi: 10.1038/nature17953, 2016.

Kürten, A., et al.: Performance of a corona ion source for measurement of sulfuric acid by chemical ionization mass spectrometry, *Atmos. Meas. Tech.*, 4, 437–443, doi: 10.5194/amt-4-437-2011, 2011.

Kürten, A., et al.: Calibration of a chemical ionization mass spectrometer for the measurement of gaseous sulfuric acid, *J. Phys. Chem. A*, 116(24), 6375–6386, doi: 10.1021/jp212123n, 2012.

Kürten, A., et al.: Neutral molecular cluster formation of sulfuric acid-dimethylamine observed in real-time under atmospheric conditions, *P. Natl. Acad. Sci. USA*, 111, 15019–15024, doi: 10.1073/pnas.1404853111, 2014.

Kürten, A., et al.: On the derivation of particle nucleation rates from experimental formation rates, *Atmos. Chem. Phys.*, 15, 4063–4075, doi: 10.5194/acp-15-4063-2015, 2015a.

Kürten, A., et al.: Observation of new particle formation and measurement of sulfuric acid, ammonia, amines and highly oxidized organic molecules at a rural site in central Germany, *Atmos. Chem. Phys.*, 16, 12793–12813, doi: 10.5194/acp-16-12793-2016, 2016b.

Lehtipalo, K., et al.: The effect of acid–base clustering and ions on the growth of atmospheric nano-particles, *Nature Commun.*, 7, 11594, doi: 10.1038/ncomms11594, 2016.

Mäkelä, J. M., et al.: Chemical composition of aerosol during particle formation events in boreal forest, *Tellus*, 53B, 380–393, doi: 10.1034/j.1600-0889.2001.530405.x, 2001.

Olenius, T., et al.: New particle formation from sulfuric acid and amines: Comparison of monomethylamine, dimethylamine, and trimethylamine, *J. Geophys. Res. Atmos.*, 122, 7103–7118, doi: 10.1002/2017JD026501, 2017.

Praplan, A. P., et al.: Dimethylamine and ammonia measurements with ion chromatography during the CLOUD4 campaign, *Atmos. Meas. Tech.*, 5, 2161–2167, doi: 10.5194/amt-5-2161-2012, 2012.

Simon, M., et al.: Detection of dimethylamine in the low pptv range using nitrate chemical ionization atmospheric pressure interface time-of-flight (CI-APi-TOF) mass spectrometry, *Atmos. Meas. Tech.*, 9, 2135–2145, doi: 10.5194/amt-9-2135-2016, 2016.

Sipilä, M., et al.: Bisulfate – cluster based atmospheric pressure chemical ionization mass spectrometer for high-sensitivity (< 100 ppqV) detection of atmospheric dimethyl amine: proof-of-concept and first ambient data from boreal forest, *Atmos. Meas. Tech.*, 8, 4001–4011, doi: 10.5194/amt-8-4001-2015, 2015.

Vaittinen, O., et al.: Adsorption of ammonia on treated stainless steel and polymer surfaces, *Appl. Phys. B*, 115(2), 185–196, doi: 10.1007/s00340-013-5590-3, 2014.

Yao, L., et al.: Detection of atmospheric gaseous amines and amides by a high-resolution time-of-flight chemical ionization mass spectrometer with protonated ethanol reagent ions, *Atmos. Chem. Phys.*, 16, 14527–14543, doi: 10.5194/acp-16-14527-2016, 2016.

You, Y., et al.: Atmospheric amines and ammonia measured with a chemical ionization mass spectrometer (CIMS), *Atmos. Chem. Phys.*, 14, 12181–12194, doi: 10.5194/acp-14-12181-2014, 2014.

Yu, H., and Lee, S.-H.: Chemical ionisation mass spectrometry for the measurement of atmospheric amines, *Environ. Chem.*, 9, 190–201, doi: 10.1071/EN12020, 2012.

Yu, H., et al.: Effects of amines on formation of sub-3 nm particles and their subsequent growth, *Geophys. Res. Lett.*, 39, L02807, doi: 10.1029/2011GL050099, 2012.

Yu, H., et al.: Laboratory observations of temperature and humidity dependencies of nucleation and growth rates of sub-3 nm particles, *J. Geophys. Res. Atmos.*, 122, 1919–1929, doi: 10.1002/2016JD025619, 2017.

Zhao, J., et al.: Observation of neutral sulfuric acid-amine containing clusters in laboratory and ambient measurements, *Atmos. Chem. Phys.*, 11, 10823–10836, doi: 10.5194/acp-11-10823-2011, 2011.

Zollner, J. H., et al.: Sulfuric acid nucleation: power dependencies, variation with relative humidity, and effect of bases, *Atmos. Chem. Phys.*, 12, 4399–4411, doi: 10.5194/acp-12-4399-2012, 2012.

---

## Author Comment (AC2) · 24 Nov 2017

We thank the referee for the constructive comments, which are added in full below (in black font). Our replies are given in blue font directly after the comments; text that has been added to the manuscript is shown in red font.

**Anonymous Referee #2**

The study by Kürten et al. presents a revised analysis of their previous study on new particle formation of the sulfuric acid-dimethylamine-water system. The new results show that nucleation is even faster than previously thought and that the derived nucleation rates are in agreement with model simulations. Further, they show that even tiny mixing ratios of dimethylamine are sufficient to explain significant particle nucleation in the boundary layer.

The paper is well written and suitable for publication in ACP after some minor revisions. My comments are listed below:

General comments:

1) Somewhere in the paper it should be stated what the typical concentrations/mixing ratios of dimethylamine and sulfuric acid in the boundary layer are. Are the concentrations that were used in your experiment realistic?

We agree with the referee and have added the following to the end of section 1:

"The reanalyzed data cover sulfuric acid concentrations from ca. $1\times10^6$ to $3\times10^7$ cm$^{-3}$, which fall into the range for most observations of atmospheric boundary layer new particle formation events (e.g. Kulmala et al., 2013). The dimethylamine mixing ratio for most of the data shown in this study is ~40 pptv ($1\times10^9$ cm$^{-3}$), which is within the rather wide range of observations (0.1 to 157 pptv, i.e., $2.5\times10^6$ to $4\times10^9$ cm$^{-3}$) for C2-amines to which dimethylamine belongs to (Yao et al., 2016)."

This indicates that the concentrations of the trace gases in the present study are atmospherically relevant.

2) The difference between the Almeida et al. (2013) study and your study (same holds for the Jen et al. (2016) study) becomes not really clear from the manuscript as it is written now. If I understand your study correct your experiment is the same as the one by Almeida et al. (2013) but with an improved set-up which results in a better agreement between measurement and observations. This is of course very nice, but as you write it, it sounds like "we are doing it right and Almeida et al. did it wrong" which is not correct. I am sure they did their best at the time they performed their study. Of course, with more time and more experience as well with improving knowledge previous studies can be repeated and improved. This should be discussed more clearly and fair in the paper.

The experiment in Almeida et al. (2013) and the present study is the same. The recorded data from both studies are from the CLOUD7 experiment (conducted in fall 2012). The difference between the two studies is the method that was used to analyze the new particle formation rates. In Almeida et al. (2013) the importance of self-coagulation in the small size range (here between 1.7 nm and ~3.2 nm) was not known yet. The fact that these small particles have a big impact on the loss rates of the growing small particles became only clear while knowledge about the

high cluster concentrations in the sulfuric acid-dimethylamine-system increased (Kürten et al., 2014) and their impact on particle growth rates was quantified (Lehtipalo et al., 2016). In parallel, the analytical method for taking into account cluster-cluster-collisions in the derivation of new particle formation rates was developed (Kürten et al., 2015a). This allowed us to revisit the previously conducted experiment (CLOUD7 from Almeida et al., 2013) and to re-calculate the formation rates with the new knowledge and analytical tools.

The reviewer is correct, that the analysis in Almeida et al. (2013) was based on the knowledge and analytical tools that were available at that time. In fact, many of the authors of the current study were involved also in the Almeida et al. (2013) study. As knowledge progressed, we are now in a position that allows us to revisit the previously published data and improve the analysis. The results of this analysis are self-consistent, i.e., they confirm what has been concluded earlier (Kürten et al., 2014; Lehtipalo et al., 2016).

In order to avoid the impression that the Almeida et al. (2013) study and the present one are based on different experiments, we have added the following to section 1:

"New particle formation rates as a function of the sulfuric acid concentration from CLOUD7 were previously published (Almeida et al., 2013). However, these data are re-analyzed in the present study using an advanced method that takes into account the effect of self-coagulation in the estimation of new particle formation rates (Kürten et al., 2015a)."

Regarding the suggestion to add further information about the Jen et al. study, we are referring to the reply to comment 12). Information about the method used by Almeida et al. (2013) is provided in the context of comment 9).

Specific comments:

3) P2, l36: add "mobility diameter" so that it reads "... at a mobility diameter of 1.7 nm...".

Done.

4) P2, l47: The abundances of sulfuric acid and dimethylamine are given in different units (concentrations and mixing ratio, respectively). The same unit should be used for both consequently. However, I would prefer if the amounts of both species would be given as concentration and mixing ratio.

We agree that it is sometimes confusing if the units pptv and $cm^{-3}$ are used for different trace gases. Therefore, we decided to always provide the conversion to $cm^{-3}$ in brackets when DMA mixing ratios are mentioned.

5) P4, l97 and l111: What the abbreviation CLOUD is standing for should be rather given in the introduction in l97 than in l111 of the method section.

Done.

6) P5, l144: The brackets around the reference are obsolete.

Done.

7) P5, l152: Same here, the brackets around the reference are obsolete.

Done.

8) P5, l169 and l172: Is the unit really correct? If yes, why is it cm$^{-1}$ s$^{-0.5}$?

The unit is correct. In the equation $k_w = C_w \cdot$ sqrt($D$), the value of $C_w$ is a constant that connects the square root of the diffusivity (unit cm s$^{-0.5}$) with the wall loss rate (unit s$^{-1}$). Therefore, $C_w$ needs to have the unit cm$^{-1}$ s$^{-0.5}$.

The expression in equation (2) of the manuscript originates from Crump and Seinfeld (1981). In the original publication it is shown that the wall loss rate is proportional to sqrt($k_e \cdot D$)/$L$ (where $k_e$ is the turbulent energy dissipation rate, unit s$^{-1}$, and $L$ is a characteristic length, unit cm). From this expression it follows directly that the empirical constant $C_w$ (which combines sqrt($k_e$)/$L$) has the correct unit.

9) P8, l300: Since you compare your results to Almeida et al. (2013) it would be good if you could write more about the Almeida et al. (2013) study. How were their nucleation rates derived. What are the differences between your experiment and their experiment? Were these experimentally derived or from model simulations?

As outlined in the reply to comment 2), the experiment in Almeida et al. (2013) and in the present study is identical.

The NPF rates from Almeida et al. (2013) were not derived with a model but with equations that can be found in the literature (e.g., in Kürten et al., 2016a, JGR).

The method from Almeida et al. (2013) is now summarized at the end of section 2.3:

"The method introduced here explicitly takes into account losses that occur between particles with $d_{p1}$ and $d_{p2}$ (self-coagulation). These losses have not been taken into account by Almeida et al. (2013). Almeida et al. (2013) derived $J_{3.2nm}$ from CPC and SMPS measurements by including the corrections for wall loss, dilution and coagulation above 3.2 nm (see also Kürten et al., 2016a). However, the extrapolation to 1.7 nm was made by using the Kerminen and Kulmala equation (Kerminen and Kulmala, 2002), which does not include the effect of self-coagulation. For the system of sulfuric acid and dimethylamine, where a significant fraction of particles reside in the small size range, this process is, however, important."

10) P10, l369: Please give to the concentration the corresponding mixing ratio.

We would like to keep the unit cm$^{-3}$ for the sulfuric acid concentration, as this is the unit that is also used in the figures and in many different other publications reporting sulfuric acid concentrations (e.g., Fiedler et al., 2005; Kuang et al., 2008; Kulmala et al., 2013; Almeida et

al., 2013; Jen et al., 2014; etc.). However, related to comment 4) we have provided the numbers in units of cm$^{-3}$ in brackets whenever a DMA mixing ratio is given.

11) P11, L402ff: Concerning the discussion on the differences between model simulation and measurements. I would say that this part could be improved. Although I agree that the agreement is very good you should also admit that the nucleation rates from the model simulation are slightly higher than the measurements which will affect the development of the size distribution. Further, it seems that the differences between the measurements and the model simulation increase with time. Furthermore, the differences are largest at in between the two modes of the size distribution. That is not discussed at all. I would assume that this is caused by an uncertainty in the model. Is the coagulation really correctly calculated? Can't you play around a little with the model and check if the differences get larger/smaller than the nucleation rate is lower/higher (assuming according lower/higher concentrations of dimethylamine and sulfuric acid)? Even larger are the difference between the aerosol volumes, but from the discussion it sounds as that the agreement is perfect which is obviously not true.

We agree with the referee that further discussion and tests regarding the sensitivity of the model would be beneficial for the manuscript. Therefore, further model calculations were performed using the following scenarios (these calculations link to the base-case simulation from Fig. 2 in the manuscript):

- Variation of the sulfuric acid concentration by ±20% to investigate the effect on the evolving size distribution (Fig. 2b).
- Negligence of van der Waals forces in the simulations; van der Waals forces enhance the collision rates by roughly a factor of 2.3 for the smallest particles (Fig. 2c).
- Assumption that the particles grow by the addition of building blocks consisting of 2 dimethylamine molecules and 1 sulfuric acid molecule (2:1 ratio between base and acid, Fig. 2c) rather than due to an equal amount of acid and base (1:1 ratio).

To illustrate the results, the previous version of Fig. 2 was replaced by a new version with four panels instead of two panels (see figure on the next page). In addition, the following discussion was added to section 3.4:

"In order to test the model sensitivity to certain variations quantitatively further simulations were performed (Fig. 2b and Fig. 2c). A variation of the steady-state sulfuric acid monomer concentration by ±20% was achieved by using different monomer production rates for the high sulfuric acid case ($P_1 = 4.17 \times 10^5$ cm$^{-3}$ s$^{-1}$) and for the low sulfuric acid case ($P_1 = 2.01 \times 10^5$ cm$^{-3}$ s$^{-1}$, Fig. 2b). This rather small variation leads to significant mismatches between the modeled and measured size distributions that is also found for the aerosol volumes (Fig. 2d).

Two further scenarios were tested with the model. First, the enhancement due to van der Waals forces were turned off. This scenario results in significantly slower growth rates and the modeled size distributions do not match the measured ones at all anymore (Fig. 2c); the same is found when comparing modeled and measured aerosol volumes (Fig. 2d). Second, the aerosol density and the molecular weight of the condensing "monomer" were changed. In the base-case simulations (Fig. 2a), the density of dimethylaminium-bisulfate is 1470 kg m$^{-3}$ and the molecular weight is 0.143 kg mol$^{-1}$ because a one to one ratio between DMA and sulfuric acid is assumed. Since full neutralization of sulfuric acid by DMA would require a 2:1-ratio between base and acid, collision-controlled nucleation of $(H_2SO_4)((CH_3)_2NH)_2$ "monomers" instead of $(H_2SO_4)((CH_3)_2NH)$ was tested. Therefore, the density was decreased by 6% to account for the density change between dimethylaminium-bisulfate and dimethylaminium-sulfate (see Qiu and

Zhang, 2011) and the molecular weight was set to 0.188 kg mol$^{-1}$. As expected, the particle growth is now slightly faster due to the additional volume added by the further DMA molecules (Fig. 2c). However, the changes are rather small and the modeled size distributions move a little further away from the measurements compared to the base case scenario (Fig. 2a)."

[Figure]

***Fig 2*** *(replacing the previous version of Fig. 2): Comparison between modeled and measured size distributions and aerosol volumes. See text for details.*

Regarding the comparison between modeled and measured aerosol volume one of the qualitative statements ("somewhat larger") was replaced by providing the number of the actual disagreement (section 3.4):

"In the beginning of the experiment the modeled aerosol volume is up to ~40% larger than the measured one, but, towards the end of the experiment (ca. 4h after its start), the volumes agree quite well. Possibly this is because the overestimated modeled particle number density at small diameters is compensated by the underestimated particle concentration in the larger size range (see Fig. 2a)."

In addition, the statement "almost perfectly represented" is replaced by "well represented" in one of the sentences in section 3.4:

"The particle growth rate is well represented by the model given the good agreement between the positions of the local maxima in the size distribution and the intersections between the size distributions and the *x*-axis."

12) P11, l440: Same here as for the Almeida et al. (2013) study, add some more information how Jen et al. (2016) derived their nucleation rates. Was it a similar experiment as the one you performed. If no, what has been done differently etc.

The comparison between our study and the study by Jen et al. (2016a) refers to the measured and simulated clusters concentrations and not to measured new particle formation rates as these were not reported by Jen et al. (2016a, 2016b).

Jen et al. (2016) developed a model that allowed them to retrieve evaporation rates for the sulfuric acid-dimethylamine system from measured cluster signals. The signals were measured at a flow reactor after a reaction time of ~3 s, where the initial sulfuric acid monomer concentrations were determined prior to the mixing with the dimethylamine.

The following information was added to section 2.5:

"Within the flow tube experiments dimethylamine was mixed into a gas flow containing a known amount of sulfuric acid monomers. The products, i.e., the sulfuric acid-dimethylamine clusters were measured after a short reaction time ($\leq$ 20 s) with a chemical ionization mass spectrometer. From the measured signals, the cluster evaporation rates were retrieved from model calculations (Jen et al., 2016a). The main differences to the CLOUD study lie within the much shorter reaction time (20 s vs. steady state in CLOUD) and in the much wider range of base to acid ratios used by Jen et al. (2016a, 2016b). This allowed them to retrieve even relatively slow evaporation rates for the sulfuric acid-dimethylamine clusters. The measured cluster/particle concentrations increased with increasing base to acid ratio, eventually approaching a plateau at a dimethylamine to acid ratio of ~1. Therefore, the high dimethylamine to acid ratio used in the CLOUD7 experiment (~ 100) can probably explain why our NPF rates are compatible with collision-controlled nucleation."

13) P11-12, l439-478: This text part definitely belongs to the method section than to the result section.

We agree with the referee, and have moved the description of the model (that includes evaporation rates) to the methods section. Therefore, a new section (section 2.5) has been created. However, in that section only a brief overview and description of the extended model is given; the formulae are provided in Appendix A.

14) P12, l481: The abbreviation DMA has not been introduced yet. I would suggest to keep writing dimethylamine throughout the manuscript. Otherwise, the text is quite difficult to read with all the abbreviations that are already used.

The abbreviation DMA was introduced and defined on page 7, line 230 (section 2.4). As the abbreviation DMA is also used in various other publications (Almeida et al., 2013; Jen et al., 2014; Jen et al., 2016a, etc.) we would like to keep it.

15) P14, l529 and l530: Please give the abundances of $H_2SO_4$ and dimethylamine as concentrations and mixing ratios.

See replies to comments 4) and 10).

16) P15, l550: I would suggest to write: "This study confirms the results derived in previous studies."

Done.

17) P14/P15, l553, l575, l576: please give the according concentrations and mixing ratios.

See replies to comments 4) and 10).

18) P15, l605: It should read "numerical model".

Done.

19) P16, l608-609: units for $M$ and $W$ missing or are these dimensionless?

These parameters are dimensionless; they were introduced by McMurry and Li (2017).

20) P30, Figure 3 caption: add DMA in brackets after dimethylamine. Give the abundances for $H_2SO_4$ and dimethylamine in both concentration and mixing ratio.

Done (addition of "DMA" to the figure caption). Regarding the concentration/mixing ratio values see replies to comments 4) and 10).

**References**

Ahlm, L., et al.: Modeling the thermodynamics and kinetics of sulfuric acid-dimethylamine-water nanoparticle growth in the CLOUD chamber, *Aerosol Sci. Technol.*, 50, 1017–1032, doi: 10.1080/02786826.2016.1223268, 2016.

Almeida, J., et al.: Molecular understanding of sulphuric acid-amine particle nucleation in the atmosphere, *Nature*, 502, 359–363, doi: 10.1038/nature12663, 2013.

Crump, J. G., and Seinfeld, J. H.: Turbulent deposition and gravitational sedimentation of an aerosol in a vessel of arbitrary shape, *J. Aerosol Sci.*, 12, 405–415, doi: 10.1016/0021-8502(81)90036-7, 1981.

Fiedler, V., et al.: The contribution of sulphuric acid to atmospheric particle formation and growth: a comparison between boundary layers in Northern and Central Europe, *Atmos. Chem. Phys.*, 5, 1773–1785, doi: 10.5194/acp-5-1773-2005, 2005.

Jen, C., et al.: Stabilization of sulfuric acid dimers by ammonia, methylamine, dimethylamine, and trimethylamine, *J. Geophys. Res.-Atmos.*, 119, 7502–7514, doi: 10.1002/2014JD021592, 2014.

Jen, C. N., et al.: Chemical ionization of clusters formed from sulfuric acid and dimethylamine or diamines, *Atmos. Chem. Phys.*, 16, 12513–12529, doi: 10.5194/acp-16-12513-2016, 2016a.

Jen, C. N., et al.: Diamine-sulfuric acid reactions are a potent source of new particle formation, *Geophys. Res. Lett.*, 43, 867–873, doi: 10.1002/2015GL066958, 2016b.

Kerminen, V.-M., and Kulmala, M.: Analytical formulae connecting the "real" and the "apparent" nucleation rate and the nuclei number concentration for atmospheric nucleation events, *J. Aerosol Sci.*, 33, 609–622, doi: 10.1016/S0021-8502(01)00194-X, 2002.

Kuang, C., et al.: Dependence of nucleation rates on sulfuric acid vapor concentration in diverse atmospheric locations, *J. Geophys. Res.-Atmos.*, 113, D10209, doi: 10.1029/2007JD009253, 2008.

Kulmala, M., et al.: Direct observations of atmospheric aerosol nucleation, *Science*, 339, 943–946, *doi:* 10.1126/science.1227385, 2013.

Kürten, A., et al.: Neutral molecular cluster formation of sulfuric acid-dimethylamine observed in real-time under atmospheric conditions, *P. Natl. Acad. Sci. USA*, 111, 15019–15024, doi: 10.1073/pnas.1404853111, 2014.

Kürten, A., et al.: On the derivation of particle nucleation rates from experimental formation rates, *Atmos. Chem. Phys.*, 15, 4063–4075, doi: 10.5194/acp-15-4063-2015, 2015a.

Kürten, A., et al.: Experimental particle formation rates spanning tropospheric sulfuric acid and ammonia abundances, ion production rates and temperatures, *J. Geophys. Res.-Atmos.*, 121, 12377–12400, doi: 10.1002/2015JD023908, 2016a.

Lehtipalo, K., et al.: The effect of acid–base clustering and ions on the growth of atmospheric nano-particles, *Nature Commun.*, 7, 11594, doi: 10.1038/ncomms11594, 2016.

McMurry, P. H., and Li, C.: The dynamic behavior of nucleating aerosols in constant reaction rate systems: Dimensional analysis and generic numerical solutions, *Aerosol Sci. Technol.*, 51, 1057–1070, doi: 10.1080/02786826.2017.1331292, 2017.

Qiu, C., and Zhang, R.: Physiochemical properties of alkylaminium sulfates: hygroscopicity, thermostability, and density, *Environ. Sci. Technol.*, 46, 4474–4480, doi: 10.1021/es3004377, 2012.

Yao, L., et al.: Detection of atmospheric gaseous amines and amides by a high-resolution time-of-flight chemical ionization mass spectrometer with protonated ethanol reagent ions, *Atmos. Chem. Phys.*, 16, 14527–14543, doi: 10.5194/acp-16-14527-2016, 2016.

---

## Author Comment (AC3) · 24 Nov 2017

We thank the referee for the constructive comments, which are added in full below (in black font). Our replies are given in blue font directly after the comments; text that has been added to the manuscript is shown in red font.

**Anonymous Referee #3**

In this study, formation rates published by Almeida et al. (2013) for ternary sulfuric acid (SA) nucleation with dimethylamine (DMA) in the CLOUD chamber are re-analyzed with a method that takes into account self-coagulation. The authors argue that particle formation rates at 1.7 nm are more than a factor of 10 higher than those reported by Almeida et al. (2013), which would imply that SA-DMA new particle formation is significant at lower DMA gas-phase concentrations than previously thought. The revised formation rates agree well with rates calculated by a kinetic aerosol model at different particle diameters. Therefore, the authors conclude that nucleation for the conditions studied here proceeds at rates that are collision-controlled.

General comments:

I think this manuscript is well written and contains some interesting results and conclusions that makes it suitable for publication in ACP. However, since the manuscript focuses mainly on a re-evaluation of particle formation rates from the paper by Almeida et al. (2013), I think more information needs to be given on the approach used by Almeida et al. for extrapolating their formation rates. I suggest that the authors add a schematic diagram or a table illustrating how 1) Almeida et al. have calculated their formation rates and 2) how the authors of the present study have calculated their formation rates. Such a diagram should also include information on what instruments were used when deriving the particle formation rates, and the necessary corrections. For instance, the authors state on lines 335-338 that Almeida et al. (2013) made an extrapolation from 3 to 1.7 nm when deriving their formation rates at 1.7 nm. How was this extrapolation done?

Furthermore, the authors of the present study use data from the smallest SMPS size channel to calculate the formation rate. As the authors admit on lines 344-345, "the smallest SMPS size channels need to be corrected by large factors to account for losses and charging probability, which introduces uncertainty". How were these corrections made, and how large were the corrections relative to the actual measured number concentrations? In addition, the authors assume on line 366 that the growth rate is independent of size which adds more uncertainty. How large are these uncertainties compared to the "error" resulting from the extrapolation method used by Almeida et al. (2013)?

Another general comment I have is related to the fact that there is another recent study focusing on nanoparticle growth for the SA-DMA system in the CLOUD chamber by Ahlm et al. (2016), where most authors of this manuscript were co-authors. In that study, model simulations and measurements with three different instruments indicated an increasing particle-phase DMA/SA molar ratio with increasing particle size due to a decreasing Kelvin-effect of DMA with increasing size, from ~1.5 to 20 nm. The results of that study appear, at least to this reviewer, to be inconsistent with the view provided in this manuscript that nucleation and growth up to ~80 nm are completely collision-controlled. I think there needs to be some explanation, or at least, discussion of this issue.

The first comment refers to the different methods used by Almeida et al. (2013) and in the present study. To add further information regarding the Almeida et al. (2013) method was requested also by reviewer 2 (comment 9). To address this comment, the following paragraph was added to section 2.3:

"The method introduced here explicitly takes into account losses that occur between particles with $d_{p1}$ and $d_{p2}$ (self-coagulation). These losses have not been taken into account by Almeida et al. (2013). Almeida et al. (2013) derived $J_{3.2nm}$ from CPC and SMPS measurements by including the corrections for wall loss, dilution and coagulation above 3.2 nm (see also Kürten et al., 2016a). However, the extrapolation to 1.7 nm was made by using the Kerminen and Kulmala equation (Kerminen and Kulmala, 2002), which does not include the effect of self-coagulation. For the system of sulfuric acid and dimethylamine, where a significant fraction of particles reside in the small size range, this process is, however, important."

We think this additional information sufficiently addresses the first part of the question and would therefore not like to add another figure to illustrate the methods.
* * *
SMPS measurements and uncertainties:

The second part of the comment addresses the SMPS measurements and the uncertainties in the Almeida et al. (2013) and the present study (especially related to the uncertainty in the growth rate).

The SMPS measurements, including the necessary corrections, are further described in the context of comment 2 (see further below).

When discussing uncertainties and errors it is important to note that Almeida et al. (2013) neglected an important process in their derivation of new particle formation rates. When particles grow from small sizes to larger sizes they are subject to several loss processes. For a chamber experiment such as CLOUD three loss processes are important: 1) coagulation, 2) wall loss, and 3) dilution. Due to these losses, the particle number concentrations (and the formation rates) decrease with particle size. Therefore, when retrieving formation rates at small diameters ($d_{p1}$) from measurements made at larger sizes ($d_{p2}$), the loss processes need to be accounted for. While Almeida et al. (2013) considered, in principle, all three loss processes; coagulation was only considered with the particles larger than $d_{p2}$. However, since a large fraction of particles reside in the size range between $d_{p1}$ and $d_{p2}$ for the sulfuric acid-dimethylamine system, their coagulation (self-coagulation) needs to be taken into account as well. The Kerminen and Kulmala equation (Kerminen and Kulmala, 2002) that was used for the correction by Almeida et al. (2013) does, however, not include this effect. This leads to a significant underestimation of the formation rates at $J_{dp1}$ ($J_{1.7nm}$). For this reason, the method from Almeida et al. (2013) could not yield accurate formation rates, which was not known, however, at the time when the analysis was performed (see also comment 2) by referee #2). The differences between the formation rates from Almeida et al. (2013) and the ones calculated with the reconstruction method (section 2.3) can be as high as a factor of 50 (see Fig. 1).

In contrast, the error on the formation rates $J_{dp1}$ ($J_{1.7nm}$) from the method in the present study is not of a systematic nature but is rather due to the uncertainties in the required parameters such as the growth rate. The error on the growth rate is ±20%. On the other hand, the growth rate size-dependency found for kinetic nucleation is relatively small in the relevant diameter range

(Kürten et al., 2015a). Therefore, the systematic error caused by this effect does not cause significant deviations. For a sulfuric acid concentration $> 2 \times 10^6$ cm$^{-3}$ and collision-controlled nucleation the size dependent growth rate leads to factor of less than 2 uncertainty (Kürten et al., 2015a), which is much smaller than the factor of 50 due to the use of an incomplete method.
* * *
Comparison to Ahlm et al. (2016):

The Ahlm et al. (2016) study showed that the small particles ($< \sim 5$ nm) grow by maintaining a 1:1 ratio between base and acid. At least that is the result of the MABNAG model that was used in their study. The APi-TOF and CI-APi-TOF measurements for the charged and neutral clusters support this assumption since the number of acid and DMA molecules do roughly match each other up to a size of ca. 2 nm (see also Almeida et al., 2013; Kürten et al., 2014; Bianchi et al., 2014). For the larger particles ($d_p > 5$ nm) and high amine mixing ratios (above ca. 40 pptv) the MABNAG model predicts base to acid ratios between 1.5 and 2, i.e., the particles are rather dimethylaminium-sulfate (2:1 ratio) than dimethylaminium-bisulfate (1:1 ratio). The Ahlm et al. (2016) simulation therefore predicts a transition from a 1:1 to a 2:1 ratio when DMA is sufficiently high and the particles reach $> 5$ nm.

The model used in the present study makes use of the assumption that a 1:1 ratio between base and acid is maintained over the full size range. In order to test how the predicted size distribution would change for a 2:1 ratio, this scenario was modeled and the results are shown in the revised version of the manuscript (new panel in Fig. 2, panel c). As expected, the results for the 2:1 ratio simulation indicate a somewhat faster growth. However, the effect is relatively small and makes the comparison between measured and simulated size distribution less good compared to the base-case scenario (1:1 ratio between base and acid). Therefore, we do not think that from the perspective of the kinetic model it makes a significant difference whether a 1:1 or a 2:1 ratio is assumed. As for the new particle formation rates, these are almost certainly better represented by the 1:1 ratio because there is direct evidence from the measurements with the mass spectrometers that the clusters and small particles maintain this ratio up to $\sim 2$ nm.

The following discussion was added to section 4:

"It is not yet clear what exact base to acid ratio the particles have for a given diameter. The clusters and small particles ($< \sim 2$ nm) seem to grow by maintaining a 1:1 ratio between base and acid, which follows from measurements using mass spectrometers (Almeida et al., 2013; Kürten et al., 2014; Bianchi et al., 2014). The larger particles could eventually reach a 2:1 ratio between base and acid, especially at the DMA mixing ratios relevant for this study (Ahlm, et al., 2016). However, even when a 2:1 ratio is assumed in the model (Fig. 2c) the expected size distributions would not change significantly compared with the base-case scenario (1:1 ratio). Therefore, it is not possible from our comparisons to find out if and at what diameter a transition from 1:1 to 2:1 base to acid ratio takes place."

Specific comments:

1. In the Almeida et al. (2013) paper, the ACDC model reproduced ternary SA-ammonia formation rates almost perfectly, but somewhat over-predicted ternary SA-DMA formation rates compared to observations in the CLOUD chamber. I think it could be worth mentioning that the conclusion within this manuscript, that ternary SA-DMA formation rates in the CLOUD

chamber were underestimated by Almeida et al., brings the formation rates much closer to predictions by the ACDC model.

We thank the referee for pointing this out. We have added the lines from the ACDC calculation in Almeida et al. (2013) to Fig. 1 for a comparison.

Furthermore, the following discussion was added to section 3.3:

"The higher formation rates are also consistent with calculations from the ACDC (Atmospheric Cluster Dynamics Code) model (McGrath et al., 2012) that were previously published in Almeida et al. (2013). Figure 1 shows the rates calculated by the ACDC model (black lines). It should be noted that these values refer to a mobility diameter of 1.2 to 1.4 nm and therefore, somewhat higher rates are expected due to the smaller diameter compared to $J_{1.7nm}$. However, the agreement between the measured and predicted rates from ACDC are now in much better agreement than before."

2. Sect. 2.1: Please describe the SMPS measurements including corrections.

The differential mobility analyzer used for the SMPS measurements is a home-built instrument with a $Kr^{85}$ neutralizer. The corrections required to retrieve the true particle number density for each of the size channels take into account a) the charging efficiency of the particles and b) the diffusion losses within the sampling lines, charger and the differential mobility analyzer as a function of the particle diameter.

The first correction (charging efficiency) yields a factor of ~50, while the second correction (transmission efficiency) requires a factor of ~3.6 for the smallest diameter (4.3 nm). The values for the charging efficiency can be determined from Wiedensohler and Fissan (1988) and the transmission can be calculated from Karlsson and Martinsson (2003) using an effective length of 8.1 m and a flow rate of 1.5 liters per minute for this SMPS system.

The following was added to section 2.1:

"The SMPS uses a differential mobility analyzer built by the Paul Scherrer Institute; it includes a $Kr^{85}$ charger to bring the particles into a charge equilibrium before they are classified. The retrieval of the particle size distributions requires corrections for the charging and the transmission efficiency, which were performed according to the literature (Wiedensohler and Fissan, 1988; Karlsson and Martinsson, 2003)."

3. Line 35: The word "advanced" is not very useful for the reader. It is better to try to explain as clearly as possible the difference between the approach used here and the method used by Almeida et al.

In principle, we agree. However, in the abstract we would not like to include too many details about the method. Therefore, the information that it is currently provided:

"…due to earlier approximations in correcting particle measurements made at larger detection threshold."

should be sufficient. However, further information about the method used by Almeida et al. (2013) was added to the end of section 2.3.

4. Line 40: "modeled and measured size distributions show good agreement". I think it should be mentioned that this was for one nucleation event that you studied in detail, unless you have analyzed other events as well.

Other events were tested as well and the comparison between model and measurement yielded similar results. The one nucleation event that was studied in further detail (Fig. 2) was chosen because it was one of the longest ones (duration of ~6h) and it was carried out at relatively high sulfuric acid concentrations. Therefore, the particles could grow to large diameters and the comparison between model and experiment covered a wide size range.

The following information was added to section 3.4:

"Comparison between modeled and measured size distributions yielded similar results for other experiments from CLOUD7. However, the experiment shown in Fig. 2 was carried out over a relatively long time (6 h) at high sulfuric acid concentrations. Therefore, the particles could grow to large diameters and the comparison between model and experiment covers a wide size range."

5. Lines 137-138: To what extent was dimethylamine oxidized by OH within the chamber during these events? Were any oxidation products detected and may these have contributed to new particle formation?

To answer this question we will first estimate the expected OH concentration during the experiments (we refer to the experiment shown in Fig. 2):

The sulfuric acid monomer production rate is

$$P_1 = k_{OH+SO2} \cdot [OH] \cdot [SO_2] \tag{1}$$

Using a value of $P_1 = 2.9 \times 10^5$ cm$^{-3}$ s$^{-1}$ (see section 3.4), $k_{OH+SO2} = 9 \times 10^{-12}$ cm$^3$ s$^{-1}$ (Atkinson et al., 2004) and $[SO_2] = 1.5 \times 10^{12}$ cm$^{-3}$ (60 ppbv of SO$_2$ were used in the experiment) the estimated OH concentration is $2.1 \times 10^4$ cm$^{-3}$.

This value of OH can be used to estimate the concentration of products from the reaction between DMA and OH (assuming steady-state conditions):

$$C_{DMA+OH \; products} = \frac{k_{OH+DMA} \cdot [OH] \cdot [DMA]}{k_w} \tag{2}$$

Here it is assumed that the products are "sticky", i.e., they are irreversibly lost to the chamber walls with the rate $k_w$. The DMA mixing ratio (concentration) is 40 pptv ($1 \times 10^9$ cm$^{-3}$), the reaction rate between OH and DMA is $6.5 \times 10^{-11}$ cm$^3$ s$^{-1}$ (Carl and Crowley, 1998) and the wall loss rate is $2 \times 10^{-3}$ s$^{-1}$. This results in a concentration of the products of $7 \times 10^5$ cm$^{-3}$.

Compared with the concentration of DMA ($1 \times 10^9$ cm$^{-3}$), the concentration of the products is less than one per mille; compared with the sulfuric acid monomer the products amount ca. 7%.

In fact, if the products would contribute to nucleation and growth, this would even lower their concentration since an additional loss term in the denominator of equation (2) would need to be included. The low concentrations of the DMA oxidation products should therefore rule out a significant contribution to aerosol nucleation and growth during CLOUD.

This is further supported by the fact that no DMA oxidation products were detected in the sulfuric acid clusters measured by the CI-APi-TOF.

In the atmosphere, [OH] can be a factor of ~100 times higher than in the current study. This can lead to higher concentrations of oxidation products of DMA. However, as these products were not observed in CLOUD we have no evidence of their impact on nucleation on growth.

6. Line 321: How high is "relatively high", and how do the authors know there was no sulfuric acid in the chamber? Do the authors think this is a general problem with using a CIMS for measuring sulfuric acid?

The sulfuric acid background was sometimes higher than $1\times10^6$ cm$^{-3}$, while it is usually in the range of $1\times10^5$ cm$^{-3}$ for the CIMS instrument. During the DMA experiments in CLOUD 7 there was, however, an instrumental problem with the CIMS, which caused the high background. The measurements made with the CI-APi-TOF verified that the high background was not real, i.e., it did not originate from the CLOUD chamber. This is now mentioned in section 3.1 and the value of the high sulfuric acid background is provided. In principle, however, the CIMS is a great instrument that measured $H_2SO_4$ reliably during many CLOUD experiments.

**References**

Ahlm, L., et al.: Modeling the thermodynamics and kinetics of sulfuric acid-dimethylamine-water nanoparticle growth in the CLOUD chamber, *Aerosol Sci. Technol.*, 50, 1017–1032, doi: 10.1080/02786826.2016.1223268, 2016.

Almeida, J., et al.: Molecular understanding of sulphuric acid-amine particle nucleation in the atmosphere, *Nature*, 502, 359–363, doi: 10.1038/nature12663, 2013.

Atkinson, R., et al.: Evaluated kinetic and photochemical data for atmospheric chemistry: Volume I – Gas phase reactions of $O_x$, $HO_x$, $NO_x$ and $SO_x$ species, *Atmos. Chem. Phys.*, 4, 1461–1738, doi: 10.1002/2015JD023868, 2004.

Bianchi, F., et al.: Insight into acid-base nucleation experiments by comparison of the chemical composition of positive, negative, and neutral clusters, *Environ. Sci. Technol.*, 48, 13675–13684, doi: 10.1021/es502380b, 2014.

Carl, S. A., and Crowley, J. N.: Sequential two (blue) photon absorption by $NO_2$ in the presence of $H_2$ as a source of OH in pulsed photolysis kinetic studies: rate constants for reaction of OH with $CH_3NH_2$, $(CH_3)_2NH$, $(CH_3)_3N$, and $C_2H_5NH_2$ at 295 K, *J. Phys. Chem. A*, 102, 8131–8141, 1998.

Karlsson, M. N. A., and Martinsson, B. G.: Methods to measure and predict the transfer function size dependence of individual DMAs, *J. Aerosol Sci.*, 34, 603–625, doi: 10.1016/S0021-8502(03)00020-X, 2003.

Kerminen, V.-M., and Kulmala, M.: Analytical formulae connecting the "real" and the "apparent" nucleation rate and the nuclei number concentration for atmospheric nucleation events, *J. Aerosol Sci.*, 33, 609–622, doi: 10.1016/S0021-8502(01)00194-X, 2002.

Kürten, A., et al.: On the derivation of particle nucleation rates from experimental formation rates, *Atmos. Chem. Phys.*, 15, 4063–4075, doi: 10.5194/acp-15-4063-2015, 2015a.

McGrath, M. J., et al.: Atmospheric Cluster Dynamics Code: a flexible method for solution of the birth-death equations, *Atmos. Chem. Phys.*, 12, 2345–2355, doi: 10.5194/acp-12-2345-2012, 2012.

Wiedensohler, A., and Fissan, H. J.: Aerosol charging in high purity gases, *J. Aerosol Sci.*, 19, 867–870, doi: 10.1016/0021-8502(88)90054-7, 1988.